# Increasing the Scope as You Learn:
# Adaptive Bayesian Optimization in Nested Subspaces

**Leonard Papenmeier**
Lund University
leonard.papenmeier@cs.lth.se

**Luigi Nardi**
Lund University, Stanford University, DBtune
luigi.nardi@cs.lth.se

**Matthias Poloczek**[*]
Amazon
San Francisco, CA 94105, USA
matpol@amazon.com

## Abstract

Recent advances have extended the scope of Bayesian optimization (BO) to expensive-to-evaluate black-box functions with dozens of dimensions, aspiring to unlock impactful applications, for example, in the life sciences, neural architecture search, and robotics. However, a closer examination reveals that the state-of-the-art methods for high-dimensional Bayesian optimization (HDBO) suffer from degrading performance as the number of dimensions increases or even risk failure if certain unverifiable assumptions are not met. This paper proposes BAxUS that leverages a novel family of nested random subspaces to adapt the space it optimizes over to the problem. This ensures high performance while removing the risk of failure, which we assert via theoretical guarantees. A comprehensive evaluation demonstrates that BAxUS achieves better results than the state-of-the-art methods for a broad set of applications.

## 1 Introduction

The optimization of expensive-to-evaluate black-box functions where no derivative information is available has found many applications, for example in chemical engineering [11, 28, 31, 58, 60], materials science [23, 29, 30, 32, 52, 64, 68], aerospace engineering [39, 43], hyperparameter optimization [5, 33, 38, 61], neural architecture search [36, 57], vehicle design [14, 34], hardware design [19, 49], drug discovery [51], robotics [12, 13, 41, 46, 54], and the life sciences [18, 59, 65]. Here increasing the number of dimensions (or parameters) of the optimization problem usually allows for better solutions. For example, by exposing more process parameters of a chemical reaction, we obtain a more granular control of the process; for a design task, we may optimize a larger number of design decisions jointly; in robotics, we gain access to more sophisticated control policies.

A series of breakthroughs have recently pushed the envelope of high-dimensional Bayesian optimization and facilitated a wider adoption in science and engineering. The key challenge for further scaling is the so-called curse of dimensionality. The complexity of the task of finding an optimum grows exponentially with the number of dimensions [7, 22]. Recently, methods that rely on local 'trust regions' have gained popularity. They usually achieve good performance for problems with up to a couple of dozen input parameters. However, we observe that their performance degrades for higher-dimensional problems. This is not surprising, given that trust regions have a smaller volume but still the full dimensionality of the problem. Other state-of-the-art methods suppose the existence

---

[*]The work was done before Matthias joined Amazon.

of a low-dimensional active subspace and enjoy great scalability if they find such a space. The caveat is that its existence is usually not known for practical applications. Moreover, the user needs to 'guess' a good upper bound on its dimensionality to enjoy a good sample efficiency.

In this work, we propose a theoretically-founded approach for high-dimensional Bayesian optimization, BAxUS (**B**ayesian optimization with **a**daptively e**x**panding s**u**bspace**s**), that reliably achieves a high performance on a comprehensive set of applications. BAxUS utilizes a family of nested embedded spaces to increase the dimensionality of the domain that it optimizes over as it collects more data. As a byproduct, BAxUS can leverage an active subspace, if it exists, without requiring the user to 'guess' its dimensionality. BAxUS is based on a novel random linear subspace embedding that enables a more efficient optimization and has strong theoretical guarantees. We make the following contributions:

1. We develop BAxUS that reliably achieves excellent solutions on a broad set of high-dimensional tasks, outperforming the state-of-the-art.

2. We present a novel family of nested random embeddings that has the following properties: a) BAxUS' embedding provides a larger worst-case guarantee for containing a global optimum than the HESBO embedding proposed by [50]. b) The BAxUS embedding is an optimal *sparse embedding*, as defined in Def. 1. c) Its probability of containing an optimum converges to the one of the HESBO embedding as the input dimensionality $D \to \infty$.

3. We conduct a comprehensive evaluation on a representative collection of benchmarks that demonstrates that BAxUS outperforms the state-of-the-art methods.

The remainder of this paper is structured as follows. Section 2 states the problem and discusses related work. Section 3 presents the BAxUS algorithm and the corresponding embedding. Section 4 evaluates BAxUS on a variety of benchmarks. We give concluding remarks in Section 5.

## 2 Background

The task is to find a minimizer
$$\boldsymbol{x}^* \in \arg\min_{\boldsymbol{x} \in \mathcal{X}} f(\boldsymbol{x}),$$
where $\mathcal{X} = [-1, +1]^D$. The objective function $f : \mathcal{X} \to \mathbb{R}$ is an expensive-to-evaluate black-box function. Hence the number of function evaluations needed to find an optimizer is crucial. Evaluations may be subject to observational noise, i.e., $f(\boldsymbol{x}_i) + \varepsilon_i$, with $\varepsilon_i \sim \mathcal{N}(0, \sigma^2)$. This work focuses on *scalable high-dimensional Bayesian optimization*, where the *input dimensionality $D$* is in the hundreds, and the sampling budget may comprise a thousand or more function evaluations.

**Linear embeddings.** A successful approach for HDBO is to assume the existence of an *active subspace* [17], i.e., there exist a space $\mathcal{Z} \subseteq \mathbb{R}^{d_e}$, with $d_e \leq D$ and a function $g : \mathcal{Z} \to \mathbb{R}$, such that for all $\boldsymbol{x}$: $g(T\boldsymbol{x}) = f(\boldsymbol{x})$ where $T \in \mathbb{R}^{d_e \times D}$ is a projection matrix projecting $\boldsymbol{x}$ onto $\mathcal{Z}$ and $d_e$ is the *effective dimensionality* of the problem. In practice, both $d_e$ and $\mathcal{Z}$ are unknown.

REMBO (Random embedding BO) [71] and HESBO (Hashing-enhanced subspace BO) [50] try to capture this active subspace by a randomly chosen linear subspace. Therefore, they generate a random projection matrix $S^\mathsf{T}$ that maps from a $d$-dimensional subspace $\mathcal{Y} \in \mathbb{R}^d$ with $d \ll D$ (the *target space*) to $\mathcal{X}$. We call $d$ the *target dimensionality*. For REMBO, each entry in $S^\mathsf{T}$ is normally distributed. REMBO uses a heuristic to determine a hyperrectangle in $\mathcal{Y}$ that it optimizes over. Note that the bounded domain may not contain a point that maps to an optimizer of $f$, a risk aggravated by distortions introduced by the projection. [6, 8, 9] proposed ideas to mitigate the issue. HESBO's random projection assigns one target dimension and sign ($\pm 1$) to each input dimension. This embedding is inspired by the count-sketch algorithm [15] for estimating frequent items in data streams. The sparse projection matrix $S^\mathsf{T}$ is binary except for the signs, and each row has exactly one non-zero entry. Even though this embedding avoids REMBO's distortions, as the authors proved, it has a lower probability of containing the optimum [40]. ALEBO [40] uses a Mahalanobis kernel and imposes linear constraints on the acquisition function to avoid projecting outside of $\mathcal{X}$.

**Non-linear embeddings.** Several works use autoencoders to learn non-linear spaces for optimization, trading in sample efficiency. Tripp et al. [67] change the training objective of a variational

autoencoder (VAE) [37] to make the target space more suitable for optimization. They give higher weight to better-performing points when training the variational autoencoder (VAE) and show that this improves optimization. Moriconi et al. [47] incorporate the training of an autoencoder directly into the likelihood maximization of a Gaussian process (GP) surrogate. The computational cost is cubic in the number of samples and the input dimension. Lu et al. [42] and Maus et al. [45] use autoencoders to learn embeddings of highly structured input spaces such as kernels or molecules. Other approaches include partial least squares [10] or sliced inverse regression [16].

**High-dimensional BO in the input space.** A popular approach to make HDBO in the input space feasible is trust regions (TRs) [22, 53, 55, 75]. The TuRBO algorithm [22] optimizes over bounded TRs instead of the global space, adapting their side lengths and the center points during the optimization process. By restricting function evaluations to trust regions, TuRBO addressed the problem of over-exploration; see [22] for details. Note that the TRs have full input dimensionality, which may impact TuRBO's ability to scale to very large dimensions. Nonetheless, TuRBO set a new state-of-the-art by scaling to dozens on input dimensions and thousands of function evaluations. Wan et al. [69] extended the idea of TRs to categorical and mixed spaces by using the Hamming distance to define the TR boundaries. SAASBO [20] uses sparse priors on the GP length scales which seems particularly valuable if the active subspace is axis-aligned. Indeed, SAASBO can outperform TuRBO on certain benchmarks [20]. The cost of inference scales cubically with the number of function evaluations; thus, SAASBO is not expected to scale beyond small sampling budgets, which is confirmed by our experiments. Another line of research relies on the assumption that the input space has an additive structure [24, 35, 48, 72]. Additive GPs rely on computationally expensive sampling methods to learn a decomposition of the input variables, which limits the scalability of such methods to problems of moderate dimensionalities and sampling budgets [22, 50]. Wang et al. [70] combined the meta-level algorithm LA-MCTS with TuRBO to improve optimization performance by learning a hierarchical space partition.

In Sect. 4 we evaluate the performances of TuRBO, SAASBO, Alebo, and HeSBO. Moreover, we study the popular CMA-ES [26] and random search [4].

## 3 The BAxUS algorithm

Wang et al. [71] showed that the REMBO embedding contains the optimum in the target space with probability one if $d \geq d_e$ and if there are no bounds on the target and input spaces, i.e., $\mathcal{Y} = \mathbb{R}^d$ and $\mathcal{X} = \mathbb{R}^D$. For $d < d_e$, it is generally impossible to represent an optimum in $\mathcal{X}$ for arbitrary $f$ because $S$ projects to a $d$-dimensional subspace in $\mathcal{X}$. We call the probability of a target space to contain the optimum the *success probability*. For $d \geq d_e$, there is a positive success probability that increases with $d$ [40, 71]. The main problem is to set $d$ as small as possible to avoid the detrimental effects of the curse of dimensionality, while keeping it as large as necessary to achieve a high probability for $\mathcal{Y}$ containing an optimum.

In practice, the active subspace and its dimensionality are usually unknown. The performance of methods such as REMBO [71], HeSBO [50], and Alebo [40] depends on choosing $d$ such that the success probability is high. Therefore, they implicitly rely on guessing the effective dimensionality $d_e$ appropriately. We argue that choosing the target dimensionality is problematic in many practical applications. If chosen too small, the subspace cannot represent $f$ sufficiently well. If it is chosen too large, the curse of dimensionality slows down the optimization.

The proposed algorithm, BAxUS, operates on target spaces of increasing dimensionality while preserving previous observations. Let $d_{\text{init}}$ be the initial target dimensionality and $m$ the total evaluation budget. BAxUS starts with a $d_{\text{init}}$-dimensional embedding that is increased over time until, after $m_D \leq m$ evaluations, it roughly reaches the input dimensionality $D$. With this strategy, we can leverage the efficiency of BO in low-dimensional spaces while guaranteeing to find an optimum in the limit. Increasing the target dimensionality is enabled by a novel embedding, which lets us carry over observations from previous, lower-dimensional target spaces into more high-dimensional target spaces. We further use a TR-based approach based on Eriksson et al. [22] to carry out optimization for high target dimensions effectively. BAxUS uses a GP surrogate [73] to model the function in the target space. Algorithm 1 gives the pseudocode for BAxUS. In Appendix B, we prove global convergence for BAxUS. We will now present the different components in detail.

**Algorithm 1** BAXUS

---

**Input:** $b$: new bins per dimension split, $D$: input dimension, $m_D$: # evaluations by which the input dimension is reached.

**Output:** minimizer $\boldsymbol{x}^* \in \arg\min_{\boldsymbol{x} \in \mathcal{X}} f(\boldsymbol{x})$.

1: $d_0 \leftarrow$ initial target dimensionality of the subspace given by Eq. (3).
2: Compute initial projection matrix $S^\intercal : \mathcal{Y} \rightarrow \mathcal{X}$ by BAXUS embedding for $D$ and $d_0$.
3: Sample initial data $\mathcal{D} = \{(\boldsymbol{y}_1, f(S^\intercal \boldsymbol{y}_1)), \ldots\}$ and fit the GP surrogate.
4: $n \leftarrow 0$
5: **while** evaluation budget not exhausted **do**
6:      $L \leftarrow L_{\text{init}}$                                                  ▷ Initialize the trust region
7:      Calculate number of accepted "failures" as described in Sec. 3.4: $\tau_{\text{fail}}^s \leftarrow \max\left(1, \min\left(\left\lfloor \frac{m_i^s}{k} \right\rfloor, d_n\right)\right)$
8:      **while** $L > L_{\text{min}}$ **and** evaluation budget not exhausted **do**
9:          Find $\boldsymbol{y}$ by Thompson sampling in TR, evaluate $f(S^\intercal \boldsymbol{y})$, and add to $\mathcal{D}$.
10:          Re-fit the GP hyperparameters.
11:          Adjust trust region, see Section 3.2 for details.
12:      **if** $d_n < D$ **then**
13:          $d_{n+1} \leftarrow \min(d_n \cdot (b+1), D)$.
14:          Increase $S^\intercal$ by Algorithm 2.                              ▷ See Appendix D
15:      **else**
16:          Re-sample initial data and discard previous observations, $d_{n+1} \leftarrow d_n$.
17:      $n \leftarrow n + 1$
18: **Return** $S^\intercal(\arg\min_{\boldsymbol{y} \in \mathcal{D}} f(S^\intercal \boldsymbol{y}))$ or $S^\intercal(\arg\min_{\boldsymbol{y} \in \mathcal{D}} \mathbb{E}_n[f(S^\intercal \boldsymbol{y})])$.      ▷ Return the best observation in case of observations without noise, or the best point according to posterior mean in case of noisy observations.

---

## 3.1 The sparse BAXUS subspace embedding

The BAXUS embedding uses a sparse projection matrix to map from $\mathcal{Y}$ to $\mathcal{X}$. The number of non-zero entries in this matrix is equal to the input dimensionality $D$. Another embedding with this property is the HESBO embedding [50]. Given the $D$ and a target dimensionality $d$, HESBO samples a target dimension $\in \{1, \ldots, d\}$ and a sign $\in \{\pm 1\}$ for each input dimension uniformly at random. Conversely, each target dimension has a set of signed contributing input dimensions. We call the set of contributing input dimensions to a target dimension a *bin*. These relations implicitly define the embedding matrix $S \in \{0, \pm 1\}^{d \times D}$, where each column has exactly one non-zero entry [15]. In the HESBO embedding, the number of contributing input dimensions varies between 0 and $D$.

The interpretation of contributing input dimensions allows for an intuitive way to refine the embedding, which is shown in Figure 1. We update the embedding matrix such that contributing input dimensions of the target dimension are re-assigned to the current bin and $b$ new bins. We then say that we *split* the corresponding target dimension. Importantly, this type of embedding allows for retaining observations (see Figure 1). Assume for example, that $y_i$ is the dimension to be split. The contributing input dimensions are re-assigned to $y_i$ and three new target dimensions $y_j$, $y_k$, and $y_l$ (here, $b = 3$); the observations can be retained by copying the value of the coordinate $y_i$ to the coordinates $y_j$, $y_k$, and $y_l$. Thus, the observations are contained in the old and in the new target space. Algorithm 2 describes the procedure in detail.

In the BAXUS embedding, we force each bin of a target dimension to have roughly the same number of contributing input dimensions: the bin sizes differ by at most one. First, we create a random permutation of the input dimensions $1, \ldots, D$. The list of input dimensions is split into $\min(d, D)$ individual bins. If $d$ does not divide $D$, not all bins can have the same size. We split the permutation of input dimensions such that the $i$-th bin has size $\lceil D/d \rceil$, if $i + d \lfloor D/d \rfloor \leq D$, and $\lfloor D/d \rfloor$ otherwise. The first bins have one additional element with this construction if $d$ does not divide $D$. We further randomly assign a sign to each input dimension. The sign of the input dimensions and their assignment to target dimensions then implicitly define $S^\intercal$ (see Figure 1). We now show that the BAXUS

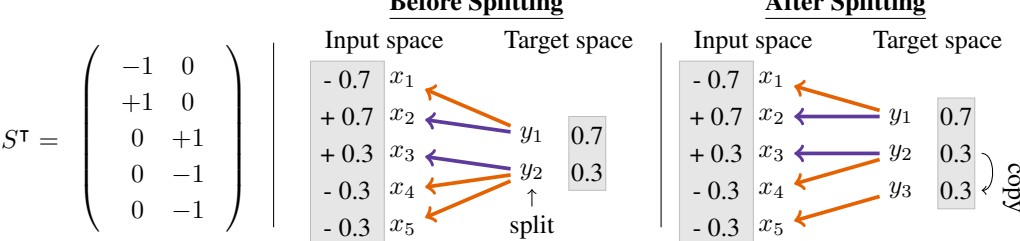

Figure 1: Observations are kept when increasing the target dimensionality. We give an example of the splitting method for $D = 5$ and $d = 2$. The first target dimension $y_1$ has two contributing input dimensions, $x_1$ and $x_2$. $y_2$ has three contributing input dimensions, $x_3$, $x_4$, and $x_5$. By $S^\mathsf{T}$, a point $(0.7, 0.3)^\mathsf{T}$ in the target space is mapped to $(-0.7, +0.7, +0.3, -0.3, -0.3)^\mathsf{T}$ in the input space. Assigning the fifth input dimension to a new target dimension and copying the function values from the second target dimension does not change the observation in the input space. The new $S^\mathsf{T}$ is not shown but has one additional column with $-1$ in the last row, and the last row of the second column is set to 0.

embedding has a strictly larger worst-case success probability than the HESBO embedding. We establish the following two definitions.

**Definition 1** (**Sparse embedding matrix**). *A matrix $S \in \{0, \pm 1\}^{d \times D}$ is a sparse embedding matrix if and only if each column in $S$ has exactly one non-zero entry* [74].

We formalize the event of "recovering an optimum" [40] as follows.

**Definition 2** (**Success of a sparse embedding**). *A success of a random sparse embedding is the event $Y^* = $ "All $d_e$ active input dimensions are mapped to distinct target dimensions."*

It is important to note that the definition of a success is sufficient but not necessary for the embedding to contain a global optimum. For example, if the origin is a global optimum, then both embeddings contain it with probability one. In that sense, the above definition provides a *worst-case guarantee*. We refer to Definition 4 in Appendix A for a formal definition of a sparse function. In Theorem 1, we give the worst-case success probability of the BAXUS embedding. All proofs have been deferred to Appendix A. Note that other than in the count-sketch algorithm [15], our hashing function is not pairwise independent. However, this does not affect our theoretical analysis.

**Theorem 1** (**Worst-case success probability of the BAXUS embedding**). *Let $D$ be the input dimensionality and $d \geq d_e$ the dimensionality of the embedding. Let $\beta_{small} = \lfloor \frac{D}{d} \rfloor$ and $\beta_{large} = \lceil \frac{D}{d} \rceil$ be the small and large bin sizes. Then the probability of $Y^*$ (see Definition 2) for the BAXUS embedding is*

$$p_B(Y^*; D, d, d_e) = \frac{\sum_{i=0}^{d_e} \binom{d(1+\beta_{small})-D}{i}\binom{D-d\beta_{small}}{d_e-i}\beta_{small}^i\beta_{large}^{d_e-i}}{\binom{D}{d_e}}. \tag{1}$$

Figure 2 shows the worst-case success probabilities of the BAXUS and HESBO embeddings for three different settings of $D$. The worst-case success probability of the HESBO embedding is given by $p_H(Y^*; d, d_e) = d!/((d - d_e)!d^{d_e})$ (see [40] and Appendix A.3). It is independent of $D$ but is shown for varying $d$-ranges on the $x$-axis, therefore the probabilities seem to change between the different subplots. Except for when $D \to \infty$, the HESBO embedding always has a non-zero failure probability, i.e., of not containing a given optimum. The BAXUS embedding ensures that the worst-case success probability is one for $d = D$. Discontinuities in the curve of the BAXUS embedding occur due to the unequal bin sizes in the BAXUS embedding's worst-case success probability. The difference between the two embeddings in Figure 2 is particularly striking when $d_e$ is high: for example, for $d_e = 20$ HESBO requires $d = 1000$ to reach a worst-case success probability of approximately $0.8$, whereas the BAXUS embedding has a success probability of 1 as soon as $d = D$. For finite $D$, HESBO's worst-case success probability is smaller than BAXUS' success probability.

**Corollary 1.** *For $D \to \infty$, the worst-case success probability of the BAXUS embedding is*

$$\lim_{D \to \infty} p_B(Y^*; D, d, d_e) = \frac{d!}{(d - d_e)!d^{d_e}},$$

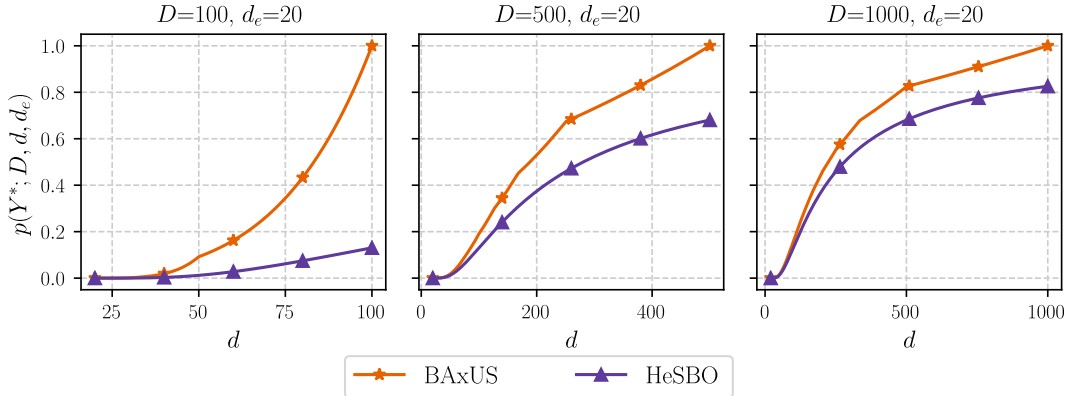

Figure 2: The worst-case guarantees for the success probabilities $p(Y^*; D, d, d_e)$ of the BAxUS and HESBO embeddings for different input dimensionalities, $D$=100 (left), $D$=500 (center), $D$=1000 (right), as a function of the target dimensionality $d$. The effective dimensionality is $d_e$=20. The BAxUS embedding has a higher worst-case success probability than the HESBO embedding. The improvement is large for input dimensionalities in the low hundreds and still substantial for 1000D tasks. In accordance with the theoretical analysis, the difference vanishes as the input dimension $D$ grows.

*and hence matches* HESBO*'s worst-case success probability $p_H(Y^*; d, d_e)$.*

We show that the BAxUS embedding is optimal among sparse embeddings.

**Corollary 2.** *With the same input, target, and effective dimensionalities ($D$, $d$, and $d_e$), no sparse embedding has a higher worst-case success probability than the* BAxUS *embedding.*

### 3.2 Trust-region approach

Similar to TURBO [22], BAxUS operates in trust regions (TRs). TRs are hyper-rectangles in the input space. Their shape is determined by their *base side length L* and the GP length scales. The side length for each dimension is proportional to the corresponding length scale of the GP kernel fitted to the data. The idea of this construction is that length scales indicate how quickly the function changes along the associated dimension. Thus, the TR is rescaled accordingly. The volume of a TR is shrunk when TURBO fails $\tau_{\text{fail}}$ consecutive times to make progress, i.e., to find a better solution. If the algorithm consecutively makes progress for $\tau_{\text{success}} = 3$ times, it expands the TR. It restarts when the base side length $L$ of the current TR falls below a threshold $L_{\min} := 2^{-7}$. In that case, it discards all observations for the TR and initializes a new TR on new samples. TRs enable TURBO to focus on regions of the space close to the incumbent, i.e., the current best solution found by the algorithm. To choose the next evaluation point, TURBO uses Thompson sampling [66], i.e., it draws a realization of the GP on a set of candidate locations in the TR and then selects a point of minimum sampled value.

TRs are an essential component of BAxUS because the target dimensionality usually grows exponentially during a run of the algorithm. We use the same hyperparameter settings as TURBO [22] with the following modifications. First, we change the criterion for when to restart a TR. Instead of restarting a TR when it becomes too small, we increase the target dimensionality by splitting each target dimension into several new bins unless BAxUS had already reached the input dimensionality. In this case, we reset the TR base side length to the initial value and re-initialize the algorithm with a new random set of initial observations. By resetting the base side length, we also avoid convergence to a particular local minimum as the TR covers large regions of the space again. TURBO solves this problem by allowing for multiple parallel TRs. Secondly, we change the number of accepted "failures" $\tau_{\text{fail}}$, such that BAxUS can roughly reach the input dimensionality in a fixed number of evaluations as described in Section 3.4.

## 3.3 Splitting strategy

Starting in a low-dimensional embedded space, BAxUS successively grows the target dimensionality to increase the probability of containing the optimum. By Corollary 2, it is optimal to keep the number of contributing input dimensions in the target bins as equal as possible. At each splitting point, BAxUS re-assigns the contributing input dimension of each target dimension. Hence the target dimensionality grows exponentially. The number of splits required to reach some input dimensionality $D$ is logarithmic in $D$. Compared to the constant growth of the target dimensionality, BAxUS uses a larger evaluation budget in each split. The algorithm starts in target dimensionality $d_{\text{init}}$. After $k$ splits, the target dimensionality is $d_k = d_{\text{init}}(b+1)^k$.

## 3.4 Controlling the number of accepted failures

BAxUS generally needs to reach high target dimensionalities to find the global optimum. As described in Section 3.2, BAxUS increases the target dimensionality when the TR base side length falls below the minimum threshold. For this to happen, the TR base length needs to be halved at least $k = \left\lfloor \log_{\frac{1}{2}} \frac{L_{\min}}{L_{\text{init}}} \right\rfloor$ times. Halving occurs if BAxUS consecutively fails $\tau_{\text{fail}}$ times in finding a better function value. If, similarly to TuRBO, we set the number of accepted "failures" $\tau_{\text{fail}}$ to the current target dimensionality of the TR, we get the lower bound $k \cdot \tau_{\text{fail}}$ on the number of function evaluations spent in that target dimensionality. This bound does not scale with the input dimensionality $D$ of the problem, i.e., the maximum target dimensionality is independent of $D$ for a fixed evaluation budget.

To enable BAxUS to reach any desired target dimensionality for the fixed evaluation budget, we scale down $\tau_{\text{fail}}$ dependent on $D$, i.e., we adjust the lower bound. We choose to make it dependent on $D$ as we are guaranteed to find the global optimum if the final target space corresponds to the input space $\mathcal{X}$ (see Appendix B). In contrast to imposing a hard limit on the number of function evaluations in a target dimensionality, scaling down the number of accepted "failures" has the advantage that we do not restrain BAxUS in cases where it finds better function values. The idea is to choose $\tau_{\text{fail}}$ dependent on the current target dimensionality $d_i$ *and* such that BAxUS can reach any desired target dimensionality. For this, we assign a minimum evaluation budget $m_i^s$ proportional to each target dimensionality $d_i$ such that $\sum_i m_i^s = m_D$ where $m_D$ is the evaluation budget until $D$ is reached.

We calculate the number of splits $n$ required to reach $D$ by

$$D \approx d_{\text{init}} \cdot (b+1)^n \Rightarrow n = \left\lfloor \log_{b+1} \frac{D}{d_{\text{init}}} \right\rceil, \tag{2}$$

with $\lfloor \cdot \rceil$ indicating rounding to the nearest integer. The minimum evaluation budget for a split is then found by multiplying $m_D$ with the "weight" of each target dimensionality: We assign each split $i$ a *split budget* $m_i^s$ that is proportional to $d_i$, such that $\sum_{i=0}^n m_i^s = m_D$, where $m_D$ is the evaluation budget until $D$ is reached:

$$m_i^s = \left\lfloor m_D \frac{d_i}{\sum_{k=0}^n d_k} \right\rceil = \left\lfloor \frac{b \cdot m_D \cdot d_i}{d_{\text{init}}((b+1)^{n+1} - 1)} \right\rceil.$$

Finally, we set the number $\tau_{\text{fail}}^i$ of accepted "failures" for the $i$-th target dimensionality $d_i$ such that (1) it just adheres to its *split budget* in the event that it never obtains a better function value, (2) it is not larger than if we would use TuRBO's choice, and (3) it is at least 1:

$$\tau_{\text{fail}}^i = \max\left(1, \min\left(\left\lfloor \frac{m_i^s}{k} \right\rfloor, d_i\right)\right).$$

**Setting the initial target dimension.** Due to the rounding in Eq. (2) and the exponential growth of the target dimensionality, the final target dimensionality $d_{\text{init}} \cdot (b+1)^n$ might differ considerably from the input dimensionality $D$. This is undesirable as we might not reach $D$ before depleting the evaluation budget, or we might overestimate the evaluation budget for the final target dimensionality $d_n$. Therefore, we set the initial target dimensionality such that the final target dimensionality is as close to $D$ as possible:

$$d_{\text{init}} = \underset{i \in \{1,\ldots,b\}}{\arg\min} |i \cdot (b+1)^n - D|, \tag{3}$$

where $n$ is given by Eq. (2). We note in passing that $1 \leq d_{\text{init}} \leq b$. An alternative to adjusting $d_{\text{init}}$ would be to fix the initial $d_0$ and adjust the growth factor $b$.

## 4 Experimental evaluation

In this section, we evaluate the performance of BAxUS on a 388D hyperparameter optimization task, a 124D design problem, and a collection of tasks that exhibit an active subspace. The BAxUS code is available at `https://github.com/LeoIV/BAxUS`.

**The experimental setup.** We benchmark against TuRBO [22] with one and five trust regions, SAASBO [20], ALEBO [40], random search [4], CMA-ES [26], and HeSBO [50], using the implementations provided by the respective authors with their settings, unless stated otherwise. For CMA-ES, we use the PyCMA [27] implementation. For HeSBO and ALEBO, we use the AX implementation [1]. To show the effect of different choices of $d$, we run HeSBO and ALEBO with $d = 10$ and $d = 20$. We observed that ALEBO and SAASBO are constrained by their high runtime and memory consumption. The available hardware allowed up to 100 evaluations for SAASBO and 500 evaluations for ALEBO for each individual run. Larger sampling budgets or higher target dimensions for ALEBO resulted in out-of-memory errors. We point out that limited scalability was expected for these two methods, whereas the other methods scaled to considerably larger budgets, as required for scalable BO. We initialize each optimizer, including BAxUS, with ten initial samples and BAxUS with $b = 3$ and $m_D = 1000$ and run 20 repeated trials. Plots show the mean performance with one standard error.

**The benchmarks.** We evaluate the selected algorithms on six benchmarks that differ considerably in their characteristics. Following [71], we augment the BRANIN2 and HARTMANN6 functions with additional dummy dimensions that have no influence on the function value. We use the 388D SVM benchmark and the 124D soft-constraint version of the MOPTA08 benchmark proposed in [20]. We set a budget of 1000 evaluations for MOPTA08, BRANIN2, and HARTMANN6 and of 2000 evaluations for the other benchmarks. Moreover, we stopped for BRANIN2 and HARTMANN6 when the simple regret dropped below .001. We show results on additional noise-free benchmarks in Appendices C.2 and C.3. We also tested the algorithms on the 300D LASSO-HIGH and the 1000D LASSO-HARD benchmarks from LASSOBENCH [59]. These benchmarks have an effective dimensionality of 5% of the input dimensionality, i.e., the LASSO-HIGH and LASSO-HARD benchmarks have 15 and 50 effective dimensions, respectively. To study the robustness to observational noise, we also tested on noisy variants of LASSO-HARD and LASSO-HIGH.

### 4.1 Experimental results

We begin with the six noise-free benchmarks. Fig. 3 summarizes the performances. On MOPTA08, a 124D vehicle design problem, SAASBO initially makes slightly faster progress than BAxUS. We suspect that this benchmark has high effective dimensionality, such that BAxUS first needs to adapt the target dimensionality to make further progress. On the 388D SVM benchmark, BAxUS adapts to the appropriate target dimensionality where it can reach good function values faster than TuRBO and CMA-ES. For this benchmark, Eriksson and Jankowiak [20] reported that SAASBO learned three active dimensions. Yet, the fact that ALEBO and HeSBO seem to stagnate after a few hundred evaluations, while BAxUS, TuRBO, and CMA-ES find better solutions, indicates that optimizing more of the 385 kernel length scales of the SVM benchmark allows for better solutions. On the 500D HARTMANN6, SAASBO performs best, closely followed by BAxUS. ALEBO and HeSBO are competitive initially but converge to suboptimal solutions. HeSBO, BAxUS, SAASBO, ALEBO all find excellent solutions on the BRANIN benchmark, with the latter algorithms converging faster.

Next, we examine the performances on the 1000D LASSO-HARD and the 300D LASSO-HIGH that exhibit active subspaces, here without observational noise. BAxUS achieves considerably better solutions than all state-of-the-art methods. We also note that TuRBO and CMA-ES perform better than SAASBO and ALEBO. While one may expect BAxUS to outperform TuRBO and CMA-ES on these tasks with high input dimensions, it is surprising that SAASBO, HeSBO, and ALEBO are not able to benefit from the present active subspace. Here BAxUS's strategy to adaptively expand the nested subspace is superior. Another crucial observation is that performances of BAxUS vary only slightly across runs. Thus, BAxUS is robust despite the stochastic construction of the

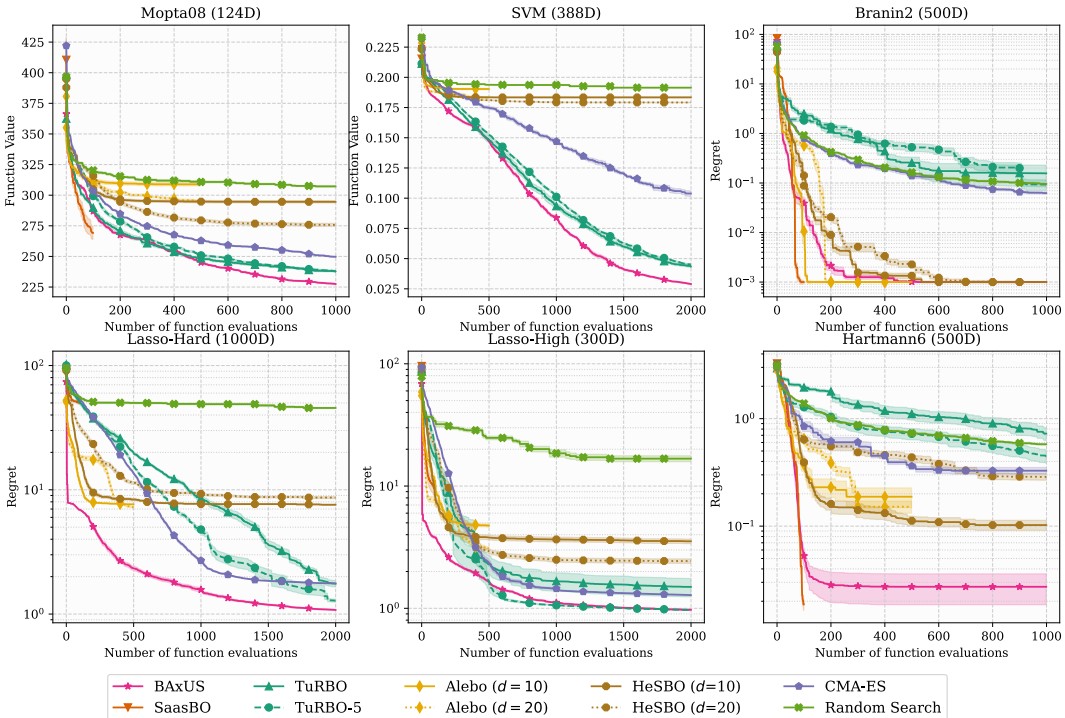

Figure 3: Top row: **124D MOPTA08 (l)**: BAXUS obtains the best solutions, followed by TURBO and CMA-ES. **388D SVM (c)**: BAXUS outperforms the other methods from the start. **500D BRANIN (r)**: SAASBO, BAXUS, ALEBO, and HESBO find an optimum; SAASBO and ALEBO converge fastest. Bottom row: **100D LASSO-HARD (l)** and **300D LASSO-HIGH (c)**: BAXUS outperforms the baselines. SAASBO, ALEBO, and HESBO struggle. **500D HARTMANN6 (r)**: SAASBO performs best, closely followed by BAXUS. The other methods show only slow progress or stagnate.

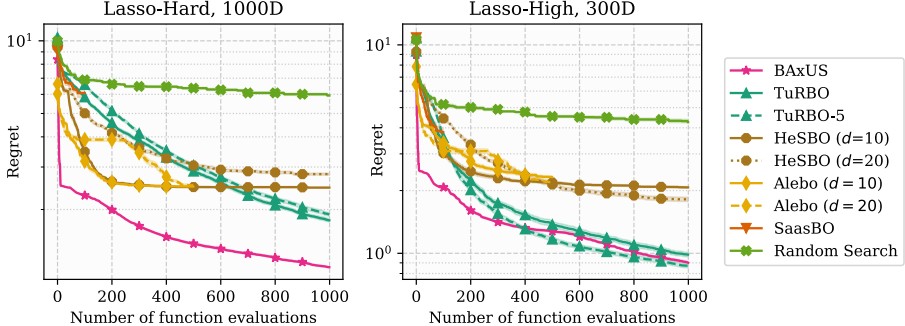

Figure 4: BAXUS outperforms the SOTA and in particular proves to be robust to observational noise on the **1000D LASSO-HARD (l)** and the **300D LASSO-HIGH (r)**. Note that CMA-ES performs considerably worse than on the noise-free versions of the benchmarks.

embedding. Across the broad collection of benchmarks, BAXUS is the only method to consistently achieve high performance.

**Noisy benchmarks.** We evaluate the algorithms also for tasks with observational noise. Fig. 4 summarizes the results. We observe that BAXUS achieves considerably better solutions for any number of observations than the competitors. Moreover, we note that the performances of SAASBO, CMA-ES, and HESBO ($d = 20$) degrade considerably on the LASSO-HIGH task compared to the noise-free formulation of the task studied above. BAXUS' performance is equally strong as for the noise-free case and keeps making progress after 1000 observations.

**BAXUS embedding ablation study.** To investigate whether the proposed family of nested random subspaces contributes to the superior performance of BAXUS, we replaced the new embedding with a similar family of nested HESBO embeddings. The results show that the proposed embedding provides a significant performance gain. Due to space constraints, the results were moved to Appendix C.1.

## 5 Discussion

High-dimensional Bayesian optimization is aspiring to unlock impactful applications broadly in science and industry. However, state-of-the-art methods suffer from limited scalability or, in some cases, require practitioners to 'guess' certain hyperparameters that critically impact the performance. This paper proposes BAXUS that works out-of-the-box and achieves considerably better performance for high-dimensional problems, as the comprehensive evaluation shows. A key idea is to scale up the dimensionality of the target subspace that the algorithm optimizes over. We apply a simple strategy that we find to work well across the board. However, we expect substantial headroom in tailoring this strategy to specific applications, either using domain expertise or a more sophisticated data-driven approach that, for example, learns a suitable target space. Moreover, future work will explore extending BAXUS to structured domains, particularly the combinatorial spaces common in materials sciences and drug discovery.

**Societal impact.** Bayesian optimization has recently become a popular tool for tasks in drug discovery [51], chemical engineering [11, 28, 31, 58, 60], materials science [23, 29, 30, 52, 64, 68], aerospace engineering [2, 39, 43], robotics [12, 13, 41, 46, 54], and many more. This speaks to the progress that Bayesian optimization has made in becoming a robust and reliable 'off-the-shelf solver.' However, this promise is not yet fulfilled for the newer field of high-dimensional Bayesian optimization that allows optimization over hundreds of 'tunable levers.' The abovementioned applications benefit from incorporating more such levers in the optimization: it allows for more detailed modeling of an aerospace design or a more granular control of a chemical reaction, to give some examples. The evaluation shows that the performance of state-of-the-art methods degenerates drastically for such high dimensions if the application does not meet specific requirements. Adding insult to the injury, such requirements as the dimensionality of an active subspace cannot be determined beforehand.

The proposed algorithm achieves a robust performance over a broad collection of tasks and thus will become a 'goto' optimizer for practitioners in other fields. Therefore, we released the BAXUS code.

## Acknowledgments and Disclosure of Funding

Luigi Nardi was supported in part by affiliate members and other supporters of the Stanford DAWN project Ant Financial, Facebook, Google, Intel, Microsoft, NEC, SAP, Teradata, and VMware. Leonard Papenmeier and Luigi Nardi were partially supported by the Wallenberg AI, Autonomous Systems and Software Program (WASP) funded by the Knut and Alice Wallenberg Foundation. Luigi Nardi was partially supported by the Wallenberg Launch Pad (WALP) grant Dnr 2021.0348. The computations were also enabled by resources provided by the Swedish National Infrastructure for Computing (SNIC) at LUNARC, partially funded by the Swedish Research Council through grant agreement no. 2018-05973. We would like to thank Erik Hellsten from Lund University and Eddy de Weerd from the University of Twente for valuable discussions.

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
