# A  Theoretical foundation for the BAxUS embedding

For convenience, we re-state Definition 1 and Definition 2 from Section 3.

**Definition 1 (Sparse embedding matrix).** A matrix $S \in \{0, \pm1\}^{d \times D}$ is a *sparse embedding matrix* if and only if each column in $S$ has exactly one non-zero entry [74].

**Definition 2 (Success of a sparse embedding).** A success of a random sparse embedding is the event $Y^* =$ "All $d_e$ active input dimensions are mapped to distinct target dimensions."

We introduce the following two definitions.

**Definition 3 (Optima-preserving sparse embedding).** A sparse embedding matrix is *optima-preserving* if each target dimension (i.e., each column in $S$) contains at most one active input dimension.

**Definition 4 (Sparse function / function with an active subspace).** Let $\mathcal{X} = [-1, 1]^D$. A function $f : \mathcal{X} \to \mathbb{R}$ has an *active subspace* (or *effective subspace* [71]), if there exist a subspace (i.e., a space $\mathcal{Z} \subseteq \mathbb{R}^{d_e}$, with $d_e \leq D$ where $d_e \in \mathbb{N}_{++}$ is the effective dimensionality and $\mathbb{N}_{++} = \mathbb{N} \setminus \{0\}$) and a projection matrix $S^\mathsf{T} \in \mathbb{R}^{D \times d_e}$, such that for any $\boldsymbol{x} \in \mathcal{X}$ there exists a $\boldsymbol{z} \in \mathcal{Z}$ so that $f(\boldsymbol{x}) = f(S^\mathsf{T}\boldsymbol{z})$ and $d_e$ is the smallest integer with this property. The function is called *sparse* if it has an active subspace and $S^\mathsf{T}$ is a sparse embedding matrix and $\mathcal{Z} = [-1, 1]^{d_e}$.

## A.1  Proof of Theorem 1

We prove the worst-case success probability for the BAxUS embedding.

**Theorem 1 (Worst-case success probability of the BAxUS embedding).** Let $D$ be the input dimensionality and $d \geq d_e$ the dimensionality of the embedding. Let $\beta_{\text{small}} = \left\lfloor \frac{D}{d} \right\rfloor$ and $\beta_{\text{large}} = \left\lceil \frac{D}{d} \right\rceil$ be the small and large bin sizes. Then the probability of $Y^*$ (see Definition 2) for the BAxUS embedding is

$$p_B(Y^*; D, d, d_e) = \frac{\sum_{i=0}^{d_e} \binom{d(1+\beta_{\text{small}})-D}{i} \binom{D-d\beta_{\text{small}}}{d_e-i} \beta_{\text{small}}^i \beta_{\text{large}}^{d_e-i}}{\binom{D}{d_e}}. \tag{4}$$

*Proof.* The assignment of input dimensions to target dimensions and the signs of the input dimensions fully define the BAxUS embedding. Note that the signs do not affect $p_B(Y^*; D, d, d_e)$ because they only correspond to "flipping" the input dimension in the target space, and our construction ensures that the value ranges are symmetric to the origin.

An assignment is optima-preserving if and only if it is possible to find a point in $\mathcal{Y}$ that maps to an optimum in $\mathcal{X}$ for any $f$. The "only if" is true because $f$ is assumed to be sparse with an active subspace with $d_e$ active dimensions. This means that the optima in $\mathcal{X}$ only change their function values along the $d_e$ active dimensions. Suppose it is possible to find a point in $\mathcal{Y}$ that maps to an arbitrary optimum in $\mathcal{X}$. In that case, the assignment is optima-preserving because it can individually adjust all the $d_e$ active dimensions in $\mathcal{X}$. However, this generally requires each active input dimension to be mapped to a distinct target dimension (note that we require being able to represent the optimum for *any* $f$). Otherwise, there would be at least two active input dimensions that cannot be changed independently. Therefore, the probability of $Y^*$ equals the probability of an optima-preserving assignment.

As all assignments are equally likely under the construction, the probability of an assignment being optima-preserving is equal to the number of possible optima-preserving assignments divided by the total number of assignments. There are $\binom{D}{d_e}$ ways of distributing the $d_e$ active dimensions across the $D$ positions, giving the denominator in Eq. (4).

Let us first assume that $\beta_{\text{small}} = \beta_{\text{large}} := \beta$, i.e., all target dimensions have the same number of input dimensions and $d$ divides $D$. We refer to this case as the *balanced case*. There are $\binom{d}{d_e}$ ways of distributing the $d_e$ active dimensions across the $d$ different target dimensions. Given one active dimension, there are $\beta$ ways in which this dimension can map to the target dimension. Therefore, for the balanced case, the worst-case success probability is given by

$$p_B(Y^*; D, d, d_e) = \frac{\beta^{d_e} \binom{d}{d_e}}{\binom{D}{d_e}}. \tag{5}$$

Next, we generalize Eq. (5) for cases where $d$ does not divide $D$. We refer to this case as the *near-balanced case*. In that case, there are two bin sizes: $\beta_{\text{small}}$ and $\beta_{\text{large}}$ with $\beta_{\text{large}} = \beta_{\text{small}} + 1$. There are $d\beta_{\text{large}} - D$ small bins (i.e., bins with bin size $\beta_{\text{small}}$) and $D - d\beta_{\text{small}}$ large bins: $D - d\beta_{\text{small}}$ gives the number of input dimensions that would not be covered if all bins were small. Since $\beta_{\text{small}}$ and $\beta_{\text{large}}$ differ by 1, this also gives the number of bins that have to be large. Conversely, if we only had large bins, we would cover $d\beta_{\text{large}} - D$ too many input dimensions. Therefore, we need $D - d\beta_{\text{small}}$ large and $d\beta_{\text{large}} - D$ small bins.

We consider all ways of distributing the $d_e$ active dimensions across the the $d\beta_{\text{large}} - D$ small and $D - d\beta_{\text{small}}$ large bins so that there is at most one active dimension in each bin. Recall that this number gives the numerator in Eq. (4). For a conflict-free assignment, if $i$ active dimensions are mapped to small bins, then $d_e - i$ active dimensions must be assigned to large bins. There are $\binom{d(1+\beta_{\text{small}})-D}{i}\binom{D-d\beta_{\text{small}}}{d_e-i}$ such assignments. Here we use that $1 + \beta_{\text{small}} = \beta_{\text{large}}$ holds for the near-balanced case. Recall that each small bin has $\beta_{\text{small}}$ locations and that each large bin has $\beta_{\text{large}}$ locations that an active dimension can be assigned to. Because $0 \leq i \leq d_e$ by construction, the number of assignments that result in an optima-preserving embedding is

$$\sum_{i=0}^{d_e} \binom{d(1+\beta_{\text{small}})-D}{i}\binom{D-d\beta_{\text{small}}}{d_e-i}\beta_{\text{small}}^i\beta_{\text{large}}^{d_e-i}.$$

Note that we leverage the facts $\binom{0}{0} = 1$, $\binom{0}{x} = 0$ for all $x \geq 1$, $\binom{y}{x} = 0$ if $x > y \geq 0$, and $\binom{x}{0} = 1$ for all $x$, thus the sum is well defined. Recall that we already showed that the denominator is $\binom{D}{d_e}$. Therefore, Eq. (4) gives the success probability in the near-balanced case.

It is easy to see that Eq. (4) is equivalent to the near-balanced formulation in Eq. (5) when $d$ divides $D$. When $d$ divides $D$, $\beta_{\text{small}} = \beta_{\text{large}} = \beta$, $d(1 + \beta_{\text{small}}) - D = d$, and $D - d\beta_{\text{small}} = 0$. Therefore, the worst-case success probability for the near-balanced case is given by

$$p_B(Y^*; D, d, d_e) = \frac{\sum_{i=0}^{d_e}\binom{d(1+\beta_{\text{small}})-D}{i}\binom{D-d\beta_{\text{small}}}{d_e-i}\beta_{\text{small}}^i\beta_{\text{large}}^{d_e-i}}{\binom{D}{d_e}}$$

$$\overset{\beta_{\text{small}}=\beta_{\text{large}}}{=} \frac{\sum_{i=0}^{d_e}\binom{d}{i}\binom{0}{d_e-i}\beta^i\beta^{d_e-i}}{\binom{D}{d_e}}$$

$$= \frac{\beta^{d_e}\sum_{i=0}^{d_e}\binom{d}{i}\binom{0}{d_e-i}}{\binom{D}{d_e}} = \frac{\beta^{d_e}\binom{d}{d_e}}{\binom{D}{d_e}}$$

where the last equality is true because the sum is zero unless $i = d_e$.

$\square$

## A.2 Proof of Corollary 2

We prove the optimality of the BAXUS embedding in terms of worst-case success probability.

**Corollary 2.** With the same input, target, and effective dimensionalities ($D$, $d$, and $d_e$), no sparse embedding has a higher worst-case success probability than the BAXUS embedding.

*Proof.* By Definition 1, an embedding matrix $S \in \{0, \pm1\}^{D\times d}$ is sparse if each row in $S$ has exactly one non-zero entry. Such an embedding can always be interpreted as disjoint sets of signed input dimensions assigned to different target dimensions: For the $n$-th input dimension, find the column with the non-zero. The respective column gives the target dimension; the entry in the matrix itself gives the sign. Conversely, each target dimension has a set of contributing input dimensions, and we call the set of input dimensions mapping to a target dimension a "bin". The sign of the input dimensions does not influence the success probability as it does not influence the ability of an embedding to contain the optimum.

We will prove that the BAXUS embedding is optimal, i.e., every other sparse embedding has a worst-case success probability that is lower or equal. We start by giving the worst-case success probability for arbitrary bin sizes.

Let $\beta_n$ be the bin size of the $n$-th bin. By Definition 2, a success is guaranteed if each bin contains at most one active input dimension. Therefore, the worst-case success probability for arbitrary bin sizes has to consider the number of cases where each bin contains at most one active input dimension and the number of bins containing one active input dimension is equal to the number of active input dimensions $d_e$. In a bin of size $\beta_i$, the active input dimension can lie in $\beta_i$ different locations. Bins not containing an active dimension do not contribute to the worst-case success probability.

We suppose w.l.o.g. that $D \geq d$. Thus, every target dimension has at least one input dimension. For each $n$ from $1$ to $d$, let the value $i_n$ indicate whether the $n$-th bin (or target dimension) contains an active dimension ($i_n = 1$) or not ($i_n = 0$). The indicator variable $\mathbb{1}_{(\sum_{n=1}^d i_n)=d_e}$ ensures that only cases where exactly $d_e$ bins contain an active input dimension are counted. Note that $\sum_{i_1=0}^{1} \sum_{i_2=0}^{1} \cdots \sum_{i_d=0}^{1} \mathbb{1}_{(\sum_{n=1}^d i_n)=d_e} = \binom{d}{d_e}$. For each case where the $d_e$ active dimensions are assigned to $d_e$ out of $d$ disjoint bins, the term $\prod_{n=1}^d \beta_n^{i_n}$ accounts for the locations in which the active dimension can lie in the $n$-th bin. Other cases do not contribute to the worst-case success probability. The exponent ensures that only bins containing an active dimension contribute to the denominator.

Then the worst-case success probability for arbitrary bin sizes is given by

$$p_{\text{general}}(Y^*; D, d, d_e) = \frac{\overbrace{\sum_{i_1=0}^{1} \sum_{i_2=0}^{1} \cdots \sum_{i_d=0}^{1} \mathbb{1}_{(\sum_{n=1}^d i_n)=d_e} \prod_{n=1}^d \beta_n^{i_n}}^{=\binom{d}{d_e}}}{\binom{D}{d_e}}, \tag{6}$$

with $\beta_n > 0$, $\sum_{n=1}^d \beta_n = D$, and $d \geq d_e$:

As in Theorem 1, the numerator of Eq. (6) gives all ways of assigning $d_e$ active dimensions to $D$ input dimensions.

We now prove that any sparse embedding has a worst-case success probability that is less or equal to the worst-case success probability of the BAxUS embedding.

Let $\beta_{\text{small}}$, $\beta_{\text{large}}$, and $p_B(Y^*; D, d, d_e)$ as in Theorem 1. Then,

$$p_{\text{general}}(Y^*; D, d, d_e) \leq p_B(Y^*; D, d, d_e) = \frac{\overbrace{\sum_{i=0}^{d_e} \binom{d(1+\beta_{\text{small}}) - D}{i} \binom{D - d\beta_{\text{small}}}{d_e - i} \beta_{\text{small}}^i \beta_{\text{large}}^{d_e - i}}^{=\binom{d}{d_e}}}{\binom{D}{d_e}}.$$

We refer the reader to the proof of Theorem 1 for an explanation of the binomial coefficients. The fact that $\sum_{i=0}^{d_e} \binom{d(1+\beta_{\text{small}})-D}{i} \binom{D-d\beta_{\text{small}}}{d_e-i} = \binom{d}{d_e}$ can be seen by noting that $(d(1 + \beta_{\text{small}}) - D) + (D - d\beta_{\text{small}}) = d$ and applying Vandermonde's convolution [25].

We will now prove that if $d$ divides $D$, then the product in the numerator of Eq. (6) is maximized if all the factors are the same, i.e., $\beta = \frac{D}{d}$. We will then show that if $d$ does not divide $D$, the integer-solution of maximal value is attained for $\beta_{\text{large}} - \beta_{\text{small}} = 1$.

**First case ($d$ divides $D$)**  We now show that the following holds for the term $\prod_{n=1}^d \beta_n^{i_n}$ in the numerator of Eq. (6): $\prod_{n=1}^d \beta_n^{i_n} \leq \beta^{d_e}$. The numerator in Eq. (6) can also be written as $e_{d_e}(\beta_1, \ldots, \beta_d)$ where

$$e_{d_e}(\beta_1, \ldots, \beta_d) = \sum_{i_1 < i_2 < \ldots < i_{d_e}} \beta_{i_1} \beta_{i_2} \ldots \beta_{i_{d_e}}$$

$$= \sum_{i_1=0}^{1} \sum_{i_2=0}^{1} \cdots \sum_{i_d=0}^{1} \mathbb{1}_{(\sum_{n=1}^d i_n)=d_e} \prod_{n=1}^d \beta_n^{i_n}$$

is the $d_e$-th elementary symmetric function of $\beta_1, \ldots, \beta_d$ [3]. Maclaurin's inequality [3] states that

$$\frac{e_1(\beta_1, \ldots, \beta_d)}{\binom{d}{1}} \geq \sqrt{\frac{e_2(\beta_1, \ldots, \beta_d)}{\binom{d}{2}}} \geq \ldots \geq \sqrt[d_e]{\frac{e_{d_e}(\beta_1, \ldots, \beta_d)}{\binom{d}{d_e}}} \geq \ldots \geq \sqrt[d]{\frac{e_d(\beta_1, \ldots, \beta_d)}{\binom{d}{d}}} \quad (7)$$

In particular,

$$\frac{e_1(\beta_1, \ldots, \beta_d)}{\binom{d}{1}} = \frac{\sum_{i=1}^{d} \beta_i}{d} = \frac{D}{d} = \beta \tag{8}$$

holds. Taking Eq. (7) and Eq. (8) to the power $d_e$ and multiplying by $\binom{d}{d_e}$, we obtain

$$\beta^{d_e} \binom{d}{d_e} \geq e_{d_e}(\beta_1, \ldots, \beta_d) = \sum_{i_1=0}^{1} \sum_{i_2=0}^{1} \cdots \sum_{i_d=0}^{1} \mathbb{1}_{(\sum_{n=1}^{d} i_n)=d_e} \prod_{n=1}^{d} \beta_n^{i_n}, \tag{9}$$

with equality if and only if $\beta_i = \beta_j$ for $i, j \in \{1, \ldots, n\}$ [3]. Therefore, the product in the numerator of Eq. (6) is maximized if all factors are equal.

**Second case ($d$ does not divide $D$)** However, if $d$ does not divide $D$, then $\beta$ is no integer which is not feasible in our setting. The $d_e$-th elementary symmetric function $e_{d_e}(\boldsymbol{\beta})$ (see Eq. (9)) is known to be *Schur-concave* if $\beta_i \geq 0$ holds for all $i$ [44]. This condition is met by $\beta$. We use the following definition of [44]: A function $f : \mathbb{R}^d \to \mathbb{R}$ is called Schur-concave if $\boldsymbol{\gamma} \prec \boldsymbol{\beta}$ implies $f(\boldsymbol{\gamma}) \geq f(\boldsymbol{\beta})$. Here, $\boldsymbol{\gamma} \prec \boldsymbol{\beta}$ means that $\boldsymbol{\beta}$ *majorizes* $\boldsymbol{\gamma}$, i.e.,

$$\sum_{i=1}^{k} \gamma_i^{\downarrow} \leq \sum_{i=1}^{k} \beta_i^{\downarrow} \quad \text{for all } k \in \{1, \ldots, d\}, \quad \text{and} \quad \sum_{i=1}^{d} \gamma_i = \sum_{i=1}^{d} \beta_i,$$

where $\boldsymbol{\gamma}^{\downarrow}$ and $\boldsymbol{\beta}^{\downarrow}$ are the vectors of all elements in $\boldsymbol{\gamma}$ and $\boldsymbol{\beta}$ in descending order [44].

We now show that there is no integer solution $\boldsymbol{\gamma}$ such that there is a near-balanced solution that majorizes $\boldsymbol{\gamma}$.

For some near-balanced assignment $\boldsymbol{\beta}$ of small and large bins to the $d$ target dimensions, consider the vector

$$\boldsymbol{\beta}^{\downarrow} = \left( \overbrace{\underbrace{\left\lceil \frac{D}{d} \right\rceil, \ldots, \left\lceil \frac{D}{d} \right\rceil}_{D - d\beta_{\text{small}} \text{ many}}}^{=\beta_{\text{large}}}, \overbrace{\underbrace{\left\lfloor \frac{D}{d} \right\rfloor, \ldots, \left\lfloor \frac{D}{d} \right\rfloor}_{d\beta_{\text{large}} - D \text{ many}}}^{=\beta_{\text{small}}} \right)$$

of bin sizes in decreasing order. For any other BAxUS embedding given by some permutation $\boldsymbol{\beta}'$ of $\boldsymbol{\beta}$, it holds that $\boldsymbol{\beta}'^{\downarrow} = \boldsymbol{\beta}^{\downarrow}$. Note that for any assignment $\boldsymbol{\gamma} = \{\gamma_1, \ldots, \gamma_d\}$ of bin sizes over the $d$ target dimensions, it has to hold that

$$\sum_{i=1}^{d} \gamma_i = D; \; \gamma_i \geq 0, \; i \in \{1, \ldots, d\}; \; \boldsymbol{\gamma} \in \mathbb{N}_+^d.$$

By assumption, since we are in the near-balanced case, $\beta_{\text{small}} = \beta_{\text{large}} - 1$.

Assume there exists an assignment of bin sizes $\boldsymbol{\gamma}$ that is not a permutation of $\boldsymbol{\beta}$ such that $\boldsymbol{\gamma} \prec \boldsymbol{\beta}$, i.e.,

$$\sum_{i=1}^{k} \gamma_i^{\downarrow} \leq \sum_{i=1}^{k} \beta_i^{\downarrow} \quad \text{for all } k \in \{1, \ldots, d\},$$

and

$$\sum_{i=1}^{d} \gamma_i = \sum_{i=1}^{d} \beta_i,$$

and

$$\exists j : \gamma_j^{\downarrow} < \beta_j^{\downarrow}.$$

Let $\kappa$ denote the (non-empty) set of such indices. Because the elements of $\beta$ and $\gamma$ both sum up to $D$, it has to hold for all $\kappa \in \kappa$ that

$$\sum_{i=1, i \neq \kappa}^{d} \gamma_i^{\downarrow} > \sum_{i=1, i \neq \kappa}^{d} \beta_i^{\downarrow}. \tag{10}$$

Remember that $\beta$ only contains elements of sizes $\beta_{\text{small}}$ and $\beta_{\text{large}}$ with $\beta_{\text{large}} = \beta_{\text{small}} - 1$. Then, Eq. (10) can only hold if either 1) $\gamma$ contains more elements of size $\beta_{\text{large}}$ than $\beta$ or 2) if it contains at least one element that is larger than $\beta_{\text{large}}$, the largest element in $\beta$.

Both cases lead to a contradiction. In the first case,

$$\sum_{i=1}^{D - d\beta_{\text{small}}+1} \gamma_i^{\downarrow} > \sum_{i=1}^{D - d\beta_{\text{small}}+1} \beta_i^{\downarrow} \Rightarrow \gamma \not\prec \beta$$

since at least the first $D - d\beta_{\text{small}} + 1$ elements of $\gamma^{\downarrow}$ are $\left\lceil \frac{D}{d} \right\rceil$ but only the first $D - d\beta_{\text{small}}$ elements of $\beta^{\downarrow}$ are $\left\lceil \frac{D}{d} \right\rceil$ and the $D - d\beta_{\text{small}} + 1$-th element of $\beta^{\downarrow}$ is $\left\lceil \frac{D}{d} \right\rceil - 1$.

In the second case, $\gamma \not\prec \beta$ because $\gamma_1^{\downarrow} > \beta_1^{\downarrow}$. It follows that no such $\gamma$ exists. Therefore, the BAXUS embedding has a maximum worst-case success probability among sparse embeddings. $\square$

### A.3 Proof of Corollary 1

**Corollary 1.** For $D \to \infty$, the worst-case success probability of the BAXUS embedding is

$$\lim_{D \to \infty} p_B(Y^*; D, d, d_e) = \frac{d!}{(d - d_e)! d^{d_e}},$$

and hence matches HESBO's worst-case success probability $p_H(Y^*; d, d_e)$.

*Proof.* By Corollary 1, the following holds for arbitrary $1 \leq d_e \leq d \leq D$ where $d, d_e, D \in \mathbb{N}_{++}$:

$$\frac{d!}{(d - d_e)! d^{d_e}} \leq p_B(Y^*; D, d, d_e),$$

because $\frac{d!}{(d - d_e)! d^{d_e}}$ is HESBO's worst-case success probability and hence less or equal to the worst-case success probability of BAXUS.

Furthermore, by the proof of Corollary 1,

$$p_B(Y^*; D, d, d_e) \leq \frac{\beta^{d_e} \binom{d}{d_e}}{\binom{D}{d_e}},$$

because $\frac{\beta^{d_e} \binom{d}{d_e}}{\binom{D}{d_e}}$ is larger or equal the worst-case success probability of any sparse embedding, among which BAXUS is the embedding with maximum worst-case success probability and $\frac{\beta^{d_e} \binom{d}{d_e}}{\binom{D}{d_e}} = p_B(Y^*; D, d, d_e)$ if and only if $\beta_{\text{small}} = \beta_{\text{large}}$, i.e., $d$ divides $D$.

In summary, we have

$$\frac{d!}{(d - d_e)! d^{d_e}} \leq p_B(Y^*; D, d, d_e) \leq \frac{\beta^{d_e} \binom{d}{d_e}}{\binom{D}{d_e}}.$$

We now show that, for fixed $d$ and $d_e$, the sequences $\frac{d!}{(d - d_e)! d^{d_e}}$ and $\frac{\beta^{d_e} \binom{d}{d_e}}{\binom{D}{d_e}}$ converge to the same point as $D \to \infty$. We consider $\lim_{\beta \to \infty}$, which is equivalent to $\lim_{D \to \infty}$ as $\beta = \frac{D}{d}$ and $d$ is fixed.

Note, that we can consider $\beta = \frac{D}{d}$ even though it is not a valid success probability when $d$ does not divide $D$, since we are only interested in bounding the true success probability. Then,

$$\lim_{\beta \to \infty} \frac{\beta^{d_e}\binom{d}{d_e}}{\binom{D}{d_e}} = \lim_{\beta \to \infty} \frac{\beta^{d_e}d!(D-d_e)!}{D!(d-d_e)!}$$

$$= \lim_{\beta \to \infty} \frac{\beta^{d_e}d!(\beta d - d_e)!}{(\beta d)!(d-d_e)!}$$

$$= \lim_{\beta \to \infty} \beta^{d_e}\frac{d!}{(d-d_e)!}\frac{(\beta d - d_e)!}{(\beta d)!}$$

Applying Stirling's approximation [25] to the numerator and the denominator of the last factor, we obtain

$$= \lim_{\beta \to \infty} \beta^{d_e}\frac{d!}{(d-d_e)!}\frac{\sqrt{2\pi(\beta d - d_e)}\left(\frac{\beta d - d_e}{e}\right)^{\beta d - d_e}}{\sqrt{2\pi\beta d}\left(\frac{\beta d}{e}\right)^{\beta d}}\frac{r(\beta d - d_e)}{r(\beta d)}$$

$$= \lim_{\beta \to \infty} \beta^{d_e}\frac{d!}{(d-d_e)!}\sqrt{\frac{\beta d - d_e}{\beta d}}e^{d_e}\frac{(\beta d - d_e)^{\beta d - d_e}}{(\beta d)^{\beta d}}\frac{r(\beta d - d_e)}{r(\beta d)}$$

$$= \lim_{\beta \to \infty} \beta^{d_e}\frac{d!}{(d-d_e)!}\sqrt{\frac{\beta d - d_e}{\beta d}}e^{d_e}\left(\frac{\beta d - d_e}{\beta d}\right)^{\beta d}\frac{1}{(\beta d - d_e)^{d_e}}\frac{r(\beta d - d_e)}{r(\beta d)}$$

$$= \lim_{\beta \to \infty} \frac{d!}{(d-d_e)!}\sqrt{\frac{\beta d - d_e}{\beta d}}e^{d_e}\left(\frac{\beta d - d_e}{\beta d}\right)^{\beta d}\frac{\beta^{d_e}}{(\beta d - d_e)^{d_e}}\frac{r(\beta d - d_e)}{r(\beta d)}$$

$$= \lim_{\beta \to \infty} \frac{d!}{(d-d_e)!}\underbrace{\sqrt{\frac{\beta d - d_e}{\beta d}}}_{\to 1}\underbrace{e^{d_e}\left(\frac{\beta d - d_e}{\beta d}\right)^{\beta d}}_{\to e^{-d_e}}\underbrace{\left(\frac{1}{d}\right)^{d_e}}_{=d^{-d_e}}\underbrace{\left(\frac{\beta d}{\beta d - d_e}\right)^{d_e}}_{\to 1}\underbrace{\frac{r(\beta d - d_e)}{r(\beta d)}}_{\to 1}$$

$$= \frac{d!}{(d-d_e)!d^{d_e}}$$

where the following holds for the error term $r(x)$ of the Stirling approximation [56]:

$$\exp\left(\frac{1}{12x+1}\right) \leq r(x) \leq \exp\left(\frac{1}{12x}\right).$$

Then, $\frac{r(\beta d - d_e)}{r(\beta d)} \to 1$ for $\beta \to \infty$ holds since

$$\frac{r(\beta d - d_e)}{r(\beta d)} \leq \exp\left(\frac{1}{12(\beta d - d_e)} - \frac{1}{12\beta d + 1}\right)$$

$$= \exp\left(\frac{12d_e + 1}{12^2\beta^2d^2 + 12\beta d - 12^2\beta dd_e - 12d_e}\right)$$

$$= \exp\left(\frac{d_e + \frac{1}{12}}{\beta d(12\beta d + 1 - 12d_e) - d_e}\right),$$

and

$$\frac{r(\beta d - d_e)}{r(\beta d)} \geq \exp\left(\frac{1}{12(\beta d - d_e) + 1} - \frac{1}{12\beta d}\right)$$

$$= \exp\left(\frac{12d_e - 1}{12^2\beta^2d^2 + 12\beta d - 12^2\beta dd_e - 12\beta d_e}\right)$$

$$= \exp\left(\frac{d_e - \frac{1}{12}}{\beta d(12\beta d + 1 - 12d_e) - d_e}\right),$$

which both go to 1 as $\beta \to \infty$.

Hence, BAxUS' worst-case success probability is bounded from below and above by sequences that converge to the same point as $D \to \infty$. The squeeze theorem (e.g., [62]) implies

$$\lim_{D \to \infty} p_B(Y^*; D, d, d_e) = \frac{d!}{(d - d_e)! d^{d_e}}.$$

$\square$

## B    Consistency of BAxUS

We prove the global convergence of function values for BAxUS. The proof idea is similar to Eriksson and Poloczek [21] but relaxes the assumption of a unique global minimizer. By construction, $f$ is sparse (see Definition 4), i.e., there exists a set of dimensions of $f$ that do not influence the function value. Thus, an optimal solution stays optimal regardless of how non-active dimensions are set. This is why we must relax the assumption of a unique global minimizer in the input space. Instead, we assume a unique global minimizer in the active subspace $z^* \in \mathcal{Z}$ that can map to arbitrarily many minimizers in the input space.

**Theorem 2** (**BAxUS consistency**). *With the following definitions:*

D1. *$\{x_k\}_{k=1}^{\infty}$ is a sequence of points of decreasing function value;*

D2. *$x^* \in \arg\min_{x \in \mathcal{X}} f(x)$ is a minimizer in $\mathcal{X}$;*

*and under the following assumptions:*

A1. *$D$ is finite;*

A2. *$f$ is observed without noise;*

A3. *$f$ is sparse and bounded in $\mathcal{X}$, i.e., $\exists C \in \mathbb{R}_{++}$ s.t. $|f(x)| < C \, \forall x \in \mathcal{X}$;*

A4. *At least one of the minimizers $x_i^*$ lies in a continuous region with positive measure;*

A5. *Once BAxUS reached the input dimensionality $D$, the initial points $\{x_i\}_{i=1}^{n_{init}}$ after each TR restart for BAxUS are chosen such that $\forall \delta \in \mathbb{R}_{++}$ and $x \in \mathcal{X}$, $\exists \nu(x, \delta) > 0$: $\mathbb{P}\left(\exists i : ||x - x_i||_2 \leq \delta\right) \geq \nu(x, \delta)$, i.e., the probability that at least one point in $\{x_i\}_{i=1}^{n_{init}}$ ends up in a ball centered at $x$ with radius $\delta$ is at least $\nu(x, \delta)$;*

*$f(x_k)$ converges to $f(x^*)$ with probability 1.*

*Proof.* We first show that BAxUS must eventually arrive at an embedding equivalent to the input space. By Assumption A1, the number of accepted "failures" (i.e., the number of times BAxUS needs to fail in finding a better solution until the TR base length is shrunk) is always finite since it is always bounded by the target dimension ($\forall i \, \tau_{\text{fail}}^i \leq d_i$) which is at most equal to $D$ ($\forall i \, d_i \leq D$). By the facts that BAxUS considers any sampled point an improvement only if it improves over the current best solution by at least some constant $\gamma \in \mathbb{R}_{++}$ and that $f$ is bounded (Assumption A3), BAxUS can only perform a finite number of function evaluations without increasing the target dimensionality of its embedding.

Once BAxUS reaches $D$, it behaves like TuRBO [22] for which Eriksson and Poloczek [21] proved global convergence assuming a unique global minimizer. For the case $d_e < D$, we notice that multiple minima in the input space occur due to non-active dimensions that do not influence the function value.

The remainder of our proof is based on the convergence theorem for global search by Solis and Wets [63], which proves convergence of function values for random search with possibly multiple minima. By considering the sequence

$$\left\{ x_i' \in \arg\min_{\hat{x} \in \{x_k\}_{k=1}^{i}} f(\hat{x}) \right\}_{i=1}^{\infty}$$

of points of decreasing function values where $\{\boldsymbol{x}_k\}_{k=1}^i$ are the observations up to the $i$-th function evaluation, Definition D1 is satisfied. Additionally, by the fact that, at each TR restart, BAXUS performs random restarts with uniform probability on $\mathcal{X}$, BAXUS satisfies the assumptions of the theorem.

The Solis and Wets [63] theorem states that for a sequence $\{\boldsymbol{x}_k\}_{k=1}^\infty$ of sampling points with $\varepsilon \in \mathbb{R}_{++}$,

$$\lim_{k \to \infty} \mathbb{P}[\boldsymbol{x}_k \in R_\varepsilon] = 1$$
$$R_\varepsilon = \{\boldsymbol{x} \in \mathcal{X} : f(\boldsymbol{x}) < \alpha + \varepsilon\}$$
$$\alpha = \inf\{t : v(\boldsymbol{x} \in \mathcal{X} : f(\boldsymbol{x}) < t) > 0\}$$

where $R_\varepsilon$ is the set of $\varepsilon$-optimal function values, $\alpha$ is the *essential infimum*, and $v$ is the Lebesgue measure. Note that the essential infimum $\alpha$ is equal to the minimum if the minimizer lies in a continuous region of positive measure, i.e., $\alpha = f(\boldsymbol{x}_i^*)$. By Assumption A4 and by letting $\varepsilon \to 0$, $f(\boldsymbol{x}_k)$ converges to $f(\boldsymbol{x}_i^*)$. $\qquad\square$

## C Additional empirical evaluations

### C.1 Ablation study for the BAXUS embedding

We conduct an ablation study to investigate the difference between the BAXUS and HESBO embeddings. We run TURBO of Eriksson et al. [22] in an embedded subspace with the two different embeddings. We use a version of ACKLEY10 (ten active dimensions, i.e., $d_e = 10$), where we shift the optimum away from the origin with a uniformly random vector $\boldsymbol{\delta} \in [-32.768, 32.768]^{d_e}$ with $\delta_i \sim \mathcal{U}(-32.768, 32.768)$. The function we optimize is then

$$f_{\text{ShiftedAckley10}}(\boldsymbol{x}) = f_{\text{Ackley10}}(\boldsymbol{x} + \boldsymbol{\delta}).$$

We adjust the boundaries of the search space such that $f_{\text{ShiftedAckley10}}$ is evaluated on the domain $\mathcal{X} = [-32.768, 32.768]^{d_e}$. The reason for shifting the optimum is that the original ACKLEY function has its optimum at the origin. In that case, any sparse embedding contains this optimum, even if all the active input dimensions are mapped to the same target dimension.

We add 10 dummy dimensions, such that $D = 30$ and set $d = 20$. With this problem-setting, the BAXUS and HESBO embeddings have a probability of approximately 0.27 and 0.07 of containing the optimum, respectively.

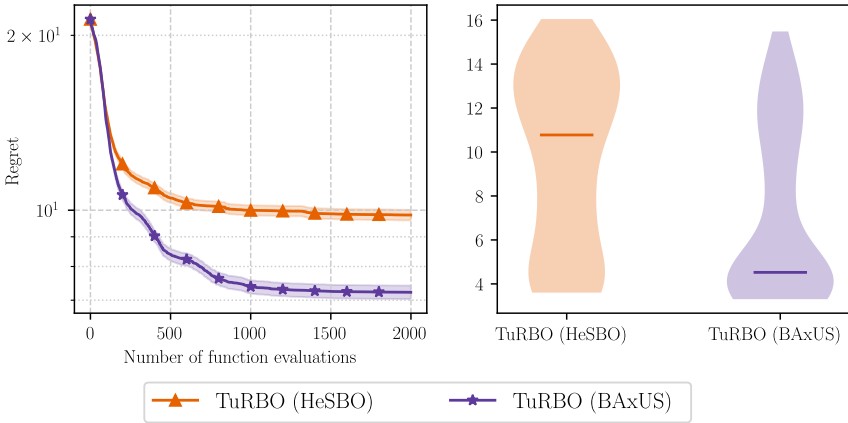

Figure 5: **Left**: the BAXUS embedding gives better optimization performance on the shifted ACK­LEY10 function: TURBO in embedded subspaces of the BAXUS and HESBO embeddings. The BAXUS embedding has a higher probability to contain the optimum. **Right**: the distribution of the final incumbents (lower the better). The horizontal bars show the median.

The left side of Figure 5 shows the incumbent mean for TURBO in the two different embedded subspaces. The shaded regions show one standard error. TURBO in an BAXUS embedding has sig­nificantly better optimization performance than in a HESBO embedding. The right side of Figure 5

shows the distributions of the final incumbents and their median. The BAxUS embedding leads to a significantly lower median and only rarely a similarly bad embedding as the HESBO method when combined with TURBO.

We perform a two-sided Wilcoxon rank-sum statistical test to check the difference between the best observed function values for the two embeddings. The difference is significant with $p \approx 0.00001$.

The performance difference between the two embeddings depends on the characteristics of the function and the different dimensionalities, the input dimensionality $D$, the target dimensionality $d$, and the effective dimensionality $d_e$. For problems with a few active dimensions and many input dimensions, the BAxUS and HESBO embeddings become more similar (see Figure 2). However, by Corollary 2, the BAxUS embedding is, in expectation and terms of success probability, always better than the HESBO embedding for arbitrary sparse functions.

For functions with an optimum at the origin, both embeddings contain that optimum regardless of $d$: Even if all active input dimensions are mapped to the same target dimension, the optimum in the input space can be reached by "setting" this particular target dimension to zero.

**TURBO with BAxUS embedding vs. BAxUS.** We compare the simple idea of running TURBO in a BAxUS embedding with the BAxUS algorithm described in Section 3. We run this simple approach for 11 different target dimensionalities $d$ $(2, 10, 20, 30, 40, 50, 60, 70, 80, 90, 100)$ on the LASSO-HARD benchmark and show the results with a sequential color map in Figure 6. Only the first $d = 2$-dimensional embedding achieves the same initial speedup as BAxUS, which is expected as BAxUS starts in a similarly low-dimensional initial embedding. However, the fixed embedding cannot explore the input space sufficiently and has the worst final solution. High-dimensional fixed embeddings have more freedom in exploring the input space; however, they suffer from slower initial optimization performance.

BAxUS has the same initial speedup as the two-dimensional fixed embedding but can explore the space further by increasing the dimensionality of its embedding.

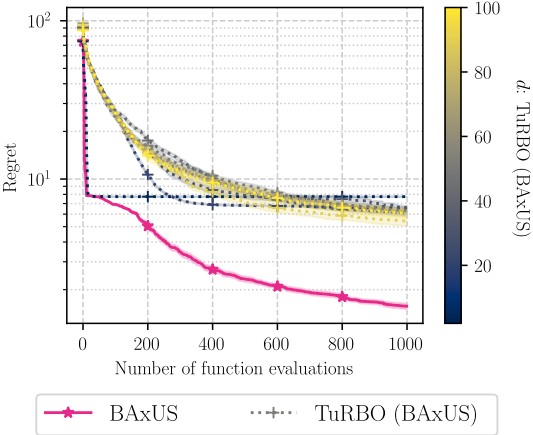

Figure 6: An evaluation of BAxUS and TURBO with BAxUS embeddings of different target dimensionalities on LASSO-HARD: We run TURBO with the BAxUS embedding for target dimensionalities $d = 2, 10, 20, 30, \ldots, 100$ and compare to BAxUS.

Summing up, we observe that BAxUS achieves a better performance than TURBO with a fixed embedding dimensionality.

## C.2 Evaluation on an additional Lasso benchmark

In addition to the synthetic LASSO-HIGH and LASSO-HARD benchmarks studied in Section 4, we evaluate BAxUS on the LASSO-DNA benchmark from LASSOBENCH [59]. The LASSO-DNA benchmark is a biomedical classification task, taking binarized DNA sequences as input [59].

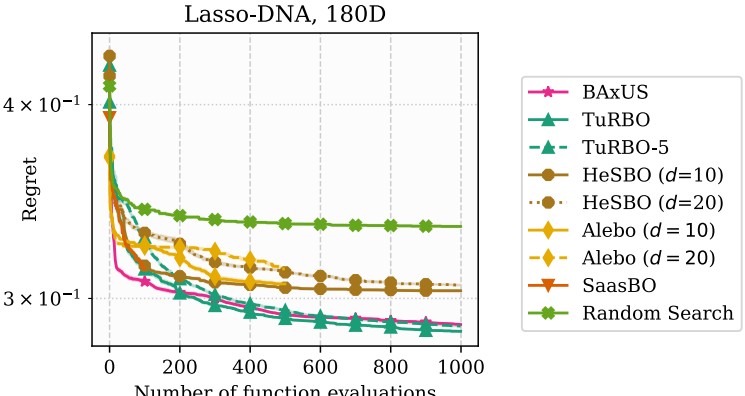

Figure 7: BAxUS and baselines on Lasso-DNA. As before, BAxUS makes considerable progress in the beginning and converges faster than TuRBO and CMA-ES.

Figure 7 shows the mean performance of BAxUS on the Lasso-DNA. Each line shows the incumbent mean; the shaded regions around the lines show one standard error. We see the same qualitative behavior as discussed in Section 4: BAxUS reaches a good initial solution faster than any other method and converges quickly.

After a worse start, TuRBO finds slightly better solutions than BAxUS.

### C.3 Evaluation on additional MuJoCo benchmarks

We evaluate BAxUS with the same baselines as in Section 4. We use the implementation of [70][2], in particular we use the `Gym` environments `Ant`, `Swimmer`, `Half-Cheetah`, `Hopper`, `Walker` 2D, and `Humanoid` 2D, all in version 2. For the 6392-dimensional `Humanoid` benchmark, we limit the target dimensionality of BAxUS to 1000 dimensions to keep the split budgets sufficiently large. For the other benchmarks, we do not limit the target dimensionality. Due to the high variance between runs, we ran all methods for 50 different runs.

We summarize the results in Fig. 8. We observe that BAxUS obtains equal or better solutions than the competitors on four out of six benchmarks. On the 120-dimensional `Walker` benchmark, BAxUS is the clear winner, followed by TuRBO and CMA-ES. On the 888-dimensional `Ant` benchmark, HeSBO finds the best solutions, followed by BAxUS that outperforms TuRBO and CMA-ES. For the 102-dimensional `Half-Cheetah`, TuRBO produces the best solutions, followed by CMA-ES and BAxUS; here, the subspace-based approaches (Alebo and HeSBO) give significantly worse solutions. For the 6392-dimensional `Humanoid` 2D, CMA-ES obtains the best solutions, followed by BAxUS, Alebo, and HeSBO.

## D  The nested family of random embeddings

We describe the method for increasing the target dimensionality under the retention of the observations. Suppose that we have collected $n$ observations and are in target dimension $d$ when Algorithm 2 is invoked. Algorithm 2 loops over the target dimensions $1, \ldots, d$. For each target dimension, the contributing input dimensions are randomly re-assigned to new bins of given sizes. This can, for example, be realized by, first, randomly permuting the list of contributing input dimensions, and, secondly, dividing the list into $b + 1$ chunks (bins). If the number of contributing input dimensions is less than $b + 1$ (remember that $b$ is the number of *new* bins), then it is not possible to re-assign the contributing input dimensions to $b + 1$ bins. Therefore, we re-assign the contributing input dimensions to $\hat{b} = \min(b, l_s - 1)$ bins, where $l_s$ is the number of contributing input dimensions to the $s$-th target dimension. This also ensures that the target dimension never grows larger than $D$ in the BAxUS embedding. We evenly distribute the $l_s$ contributing input dimensions across the $\hat{b}$ bins by

---

[2]`https://github.com/facebookresearch/LA-MCTS/blob/main/example/mujuco/functions.py`, last accessed: 06/10/2022

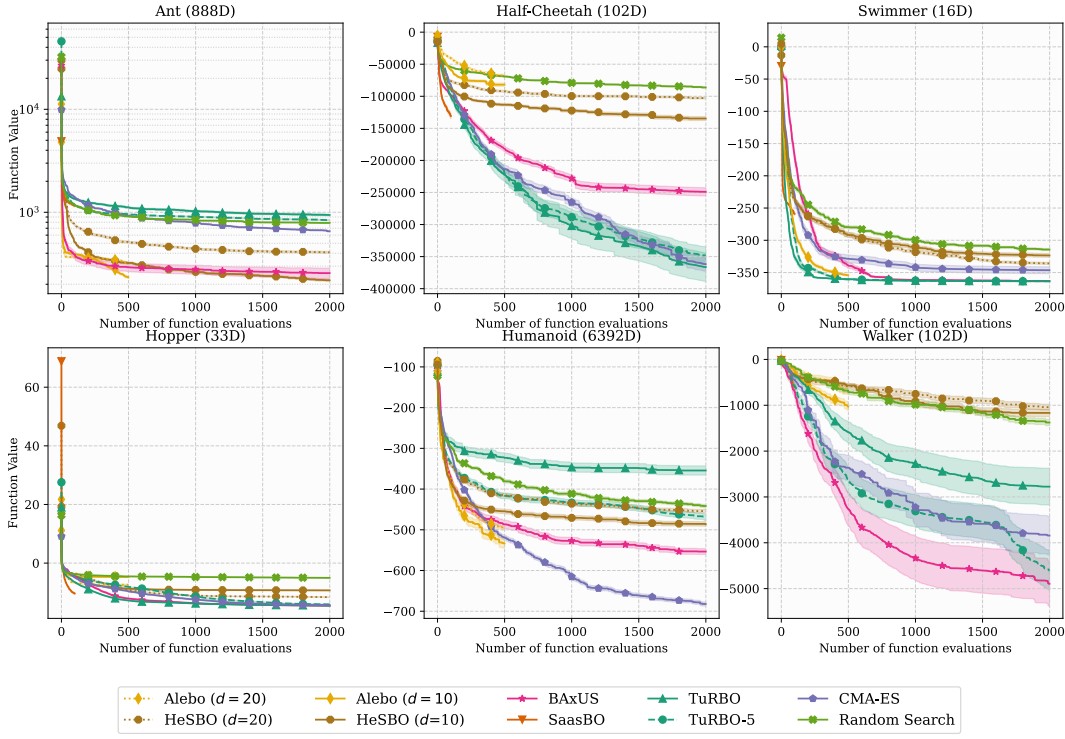

Figure 8: An evaluation of BAxUS and other methods on high-dimensional test problems of Mu-JoCo.

again using the BAxUS embedding. This gives a smaller (in terms of number of rows) projection matrix $\tilde{S}^{\mathsf{T}}$ which we finally use to update $S^{\mathsf{T}}$:

---

**Algorithm 2** Observation-preserving embedding increase

---

**Input:** transposed embedding matrix $S^{\mathsf{T}}$, number of new bins per latent dimension $b$, observed points $Y \in [-1, 1]^{n \times d}$.

**Output:** updated transposed embedding matrix $S^{\mathsf{T}}$ and updated observation matrix $Y$

  **for** $s \in \{1, \ldots, d\}$ **do**

    $\boldsymbol{D}^s \leftarrow$ contributing input dimensions of $s$-th latent dimension of the current embedding

    $l_s \leftarrow |\boldsymbol{D}^s|$

    $\hat{b} \leftarrow \min(b, l_s - 1)$                          ▷ If $l_s - 1 < b$, we can at most create $l_s - 1$ new bins.

    Copy and append $s$-th column of $Y$ $\hat{b}$ times at the end of $Y$.

    Add $\hat{b}$ zero columns at the end of $S^{\mathsf{T}}$.

    $\boldsymbol{\sigma} \leftarrow$ signs of dimensions $\in \boldsymbol{D}^s$.

    $\tilde{S}^{\mathsf{T}} \leftarrow$ Baxus-Embedding$(l_s, \hat{b} + 1)$    ▷ Re-assign input dims. equally[3], $\tilde{S}^{\mathsf{T}} \in \{0, \pm 1\}^{l_s \times \hat{b}+1}$

    **for** $i \in \{1, \ldots, l_s\}, j \in \{1, \ldots, \hat{b} + 1\}$ **do**

        **if** $\tilde{S}_{ij}^{\mathsf{T}} \neq 0$ **then**

            **if** $j > 1$ **then**               ▷ Move values that fall into new bins to end of $S^{\mathsf{T}}$.

                $S_{\boldsymbol{D}_i^s, \hat{d}-\hat{b}-1+j}^{\mathsf{T}} \leftarrow \boldsymbol{\sigma}_i$             ▷ $\hat{d}$: columns of $S^{\mathsf{T}}$

                $S_{\boldsymbol{D}_i^s, s}^{\mathsf{T}} \leftarrow 0$                    ▷ Set value in "old" column to zero.

  **Return** $S^{\mathsf{T}}$ and $Y$.

---

---

[3]Equally means that all $\hat{b} + 1$ bins have roughly the same number of contributing input dimensions. The number of contributing input dimensions to the different bins differ by at most 1.

# E Additional details on the implementation and the empirical evaluation

We benchmark against SAASBO, TURBO, HESBO, ALEBO, and CMA-ES:

- For SAASBO, we use the implementation from [20] (`https://github.com/martinjankowiak/saasbo`, license: none, last accessed: 05/09/2022).

- For TURBO, we use the implementation from [22] (`https://github.com/uber-research/TuRBO`, license: `Uber`, last accessed: 05/09/2022).

- For HESBO and ALEBO, we use the implementation from [40] (`https://github.com/facebookresearch/alebo`, license: `CC BY-NC 4.0`, last accessed: 05/09/2022).

- For the LASSO benchmarks, we use the implementation from [59] (`https://github.com/ksehic/LassoBench`, license: `MIT and BSD-3-Clause`, last accessed: 05/09/2022).

We use `GPyTorch` (version 1.8.1) to train the GP with the following setup: We place a top-hat prior on the Gaussian likelihood noise, the signal variance, and the length scales of the Matérn 5/2 ARD kernel. The interval for the noise is $[0.005, 0.2]$, for the signal variance $[0.05, 20]$, and for the lengthscales $[0.005, 10]$.

We evaluate on the synthetic BRANIN2[4] and HARTMANN6[5] functions. Since we augment the function with dummy dimensions, we use the same domain for $x_1$ and $x_2$, namely $[-5, 15]^D$ for BRANIN2 and $[0, 1]^D$ for HARTMANN6.

Similar to TURBO, we sample a $\min(100d_n, 5000)$-element Sobol sequence on which we minimize the posterior sample. To maximize the marginal log-likelihood of the GP, we sample 100 initial hyperparameter configurations. The ten best samples are further optimized using the ADAM optimizer for 50 steps.

We ran the experiments for approximately 15,000 core hours on Intel Xeon Gold 6130 CPUs provided by a compute cluster.

---

[4]See `https://www.sfu.ca/~ssurjano/branin.html`, last accessed: 05/09/2022

[5]See `https://www.sfu.ca/~ssurjano/hart6.html`, last accessed: 05/09/2022