# OpenReview forum: "Increasing the Scope as You Learn: Adaptive Bayesian Optimization in Nested Subspaces"
_NeurIPS.cc/2022/Conference — NeurIPS 2022 Accept_

### Official Review · Reviewer_355n · 2022-07-11

**Rating:** 7
**Confidence:** 3
**Soundness:** 3 good
**Presentation:** 2 fair
**Contribution:** 2 fair

**Summary:**

The paper considers Bayesian optimization for sparse, axis-aligned functions over high-dimensional spaces (hundreds of dims). The proposed solution creates a lower-dimensional space by sub-selecting dimensions from the original input space. The lower-dimensional space is searched with Thompson sampling and trust regions. The dimensionality of the lower-dimensional space is increased over the course of the proposed algorithm. The paper shows that the probability of containment of the function's optimum by the proposed algorithm's lower-dimensional space is no worse than a recent high-dimensional Bayesian optimization approach, also for sparse, axis-aligned functions (HESBO). Further, a total input evaluation budget is appropriately split among the sequence of nested lower-dimensional spaces. The paper provides results on high-dimensional datasets.

**Questions:**

* Why is REMBO not included in the experimental comparison?

* What is random in the "random embedding" Alg 2?

* To what extent can the assumption of axis-aligned function be relaxed?

**Limitations:**

The paper does cover limitations and societal impacts to some degree in the paper.

**Strengths And Weaknesses:**

Originality

* The nested lower-dimensional spaces is a neat idea. I wish this direction was explored/pushed further in the paper.

Quality

* It wasn't obvious to me why REMBO was not included in the experimental comparison as it seems to be a natural competitor in this setting.

Clarity

* The setup and algorithm description are generally clear. Some of the choices described in the splitting strategy, fail tolerance control, and initial target dimension weren't immediately intuitive. I couldn't parse the extent to which these were necessary for the proposed algorithm's effectiveness over competitors.

Significance

* The general setting of high-dimensional black-box optimization is clearly important and effective general algorithms in this setting would be valuable for the community. The assumption of the function being axis-aligned is a strong one however.

---

> ### Author Response · Authors · 2022-07-30
> **Reply to Reviewer 355n**
>
> We thank the reviewer for their helpful feedback and for appreciating our nested lower-dimensional spaces idea. We have addressed the reviewer's remarks in the newly uploaded pdf.
>
> “Why is REMBO not included in the experimental comparison?”
>
> We did not compare against REMBO because Alebo and HeSBO outperform REMBO by a wide margin [references 17, 36, 45 in our paper]. This is consistent with another recent high-dimensional BO study [reference 17 in our paper] where REMBO was omitted.
>
> “What is random in the "random embedding" Alg 2?”
>
> The randomness lies in the re-assignment of input dimensions to new bins. When re-assigning input dimensions to b+1 bins, the list of input dimensions is effectively permuted and "cut" into b+1 equally sized chunks (i.e., bins) where the bin sizes differ by at most one. The permutation of the input dimensions is random. We added this explanation in the newly uploaded paper (Appendix E) to make this clear.
>
> “To what extent can the assumption of axis-aligned function be relaxed?”
>
> We changed the term “axis-aligned function” to “a function with an axis-aligned active subspace”. While we did not change the theoretical definitions, we realized that the expression “axis-aligned function” is misleading. We do not require that every effective dimension of the function can be varied independently of the others, we allow for interactions between the effective dimensions of the function (e.g., Branin has such interaction terms). We only require that unimportant dimensions do not influence the function value. Changing this expression makes it clearer that this assumption is less strict than it sounds.
>
> Additionally, we would like to emphasize that this assumption holds in many real-world applications. For example, there is a body of literature on hyperparameter importance, e.g., “Hyperparameter Importance Across Datasets” by van Rijn and Hutter (ACM SIGKDD), “Hyperparameter optimization: a spectral approach” by Hazan et al. (ICLR 2018). Similarly, the same is true for self-tuning databases, as shown in "Automatic Database Management System Tuning Through Large-scale Machine Learning" by Aken et al. (VLDB), and in algorithm configuration, as shown in "CAVE: Configuration Assessment, Visualization and Evaluation" by Biedenkapp et al.

---

> > ### Comment · Reviewer_355n · 2022-08-08
> > **Post-rebuttal response**
> >
> > Thank you for these answers and refs on the assumption of axis-aligned subspaces!
> >
> > I upgraded my score, but kept my confidence at the same level.

---

### Official Review · Reviewer_qmcj · 2022-07-12

**Rating:** 7
**Confidence:** 5
**Soundness:** 4 excellent
**Presentation:** 4 excellent
**Contribution:** 3 good

**Summary:**

This paper proposes BaXus, which extends TRBO (trust region Bayesian optimization), by using not just trust regions but a linear embedding of varying dimensionality. The linear embedding is constructed via a count-sketch style embedding that produces effectively bins of data in each region of the input space. The dimensionality of the embedding can grow or shrink as part of the TRBO procedure, much like how in standard TRBO, the bounds of the space can change.

**Questions:**

What kind of interpolations are used for the lines in Fig. 2? They seem to not be linear?

Is it possible to construct a situation where the Baxus embedding has either worse or the exact same success probability as HesBO? I know they converge in the limit but is there another situation where they can match?

Fig. 7: why is most of the improvement from BaxUs mostly at the beginning?

L763: what is a top hat prior?

Figs. 3 and 4: why do many of the Alebo and SAASBO evaluations seem to end early? Is this due to the increased cost of MCMC fitting?

-	For example, Fig. 4 right: SAASBO seems to be cut at 100 evaluations rather than the 1000 evaluations fo the other methods. Also on hartmann500.

L260: \citet for [63].

L764: By ARD kernel, do you mean RBF ARD kernel or a Matern family?

L770: Why so many random restarts for the GP fitting? Is this done for the TRBO and/or SAASBO models as well?



**Limitations:**

Mostly yes. Negative societal impliations of BayesOpt (e.g. weapons or bioweapons design) could also be mentioned.

**Strengths And Weaknesses:**

Originality:
This approach is a pretty clever combination of TRBO and embedding based approaches for high dimensional Bayes Opt. However, it ends up being very building block based, which isn’t necessarily a bad thing.

Computing the success probability of the embedding and tying the primary idea back to well studied count sketch techniques does seem to be original to me.

Quality:

The experimental evaluation and ablation studies do seem to be performed of a reasonably high quality and the plots are quite easy to parse.

The one set of high dimensional BayesOpt benchmarks that is probably missed in the experimental evaluation is that of LaMCTS (Wang et al, ’20), which uses a classifier to construct the trust regions for TRBO. It might be too much to ask for a comparison using BaxUs inside, but it seems like a fair comparison to use against standard TRBO + LaMCTS.

-	LaMCTS also has a slightly distinct and more interesting set of benchmarks which use MuJoCo environments, which are somewhat more interesting than simply adding dimensions into test problems like Branin.

Fig. 5 is very nice to see and seems to convincingly demonstrate that the Baxus embeddings perform better downstream than the HesBO embeddings, even inside of TRBO.

Clarity:

It’s really nice that the probability of computing the optimum is computable for this approach, even if it’s a bit more complicated than the HesBO expression.

The plots and writing are overall easy to parse.

However, the error bars and the marks (e.g. triangles) on the plots are confusing to me. What are they and what do they represent?

Significance:

Better methods for high dimensional black box optimization are definitely quite significant and core to the NeuriPS community.

I would have liked to have seen a more challenging benchmark task, such as control of MuJoCo environments (see quality comment).

References:

Wang, L., Fonseca, R., and Tian, Y. (2020). Learning search space partition for black-box
optimization using monte carlo tree search. In Advances in Neural Information Processing
Systems, volume 33.

---

> ### Author Response · Authors · 2022-07-30
> **Reply to Reviewer qmcj**
>
> We thank the reviewer for their detailed feedback. We are glad that the reviewer finds the approach clever and the work original. We invite the reviewer to see the updated version of the paper that includes the request of the reviewer to increase the number of evaluations for Alebo and improves the explanation of the proofs in the appendix.
>
> "More benchmark tasks, such as MuJoCo"
>
> Thank you for the suggestion! We will look for applicable benchmarks in MuJoCo to include in the camera-ready version.
>
> "Bars and marks on the plots"
>
> The error bars indicate the standard error of the mean over the repeated trials. We use markers to make it possible to follow the legend for colorblind people and if the paper is printed in black and white.
>
> “What kind of interpolations are used for the lines in Fig. 2? They seem to not be linear?”
>
> We are interpolating linearly between two different values of d. However, we believe that what the reviewer is referring to are the “bumps” in the lines for the BAxUS embedding. These occur in cases where d divides D because the success probability is higher in cases where most of the bins are of equal size (see the AM-GM part in the proof of Corollary 2).
>
> “Is it possible to construct a situation where the Baxus embedding has either worse or the exact same success probability as HesBO?
>
> We define success probability as the probability to represent the optimum for any sparse-axis aligned function of a given effective dimensionality. With this definition, the BAxUS embedding is strictly better than the HeSBO embedding in terms of success probability. However, if you consider only a single function where the optimum lies in the origin, HeSBO and BAxUS have the same probability of 1 to contain the optimum: even if the HeSBO embedding draws the same target dimension for every input dimension, it will still be able to represent the optimum as it can be reached by setting that dimension to zero. The same holds for BAxUS.
>
> “Fig. 7: why is most of the improvement from BaxUs mostly at the beginning?”
>
> The intuition for this behavior is that BAxUS initially only needs to fit the GP on a low-dimensional problem. Exploring a lower-dimensional space is easier than exploring a high-dimensional space. After the initial progress, BAxUS roughly shows the same speed of convergence as the baselines.
>
> “L763: what is a top hat prior?”
>
> With top hat prior, we mean a uniform probability distribution in the stated range of hyperparameters.
>
> “Figs. 3 and 4: why do many of the Alebo and SAASBO evaluations seem to end early? Is this due to the increased cost of MCMC fitting?
>
> Indeed we had to limit the number of evaluations of Saasbo and Alebo due to computational constraints (see lines 253-254 in the non-revised paper). For Alebo, the Mahalanobis kernel requires extensive computation. For Saasbo, the NUTS sampler requires extensive computation.
>
> "L764: By ARD kernel, do you mean RBF ARD kernel or a Matern family?"
>
> We mean a 5/2 Matern kernel, we clarified this in the text.
>
> "L770: Why so many random restarts for the GP fitting? Is this done for the TRBO and/or SAASBO models as well?"
>
> The hyperparameter optimization procedure works by first sampling 100 random hyperparameter configurations and then starting gradient descent with Adam on the ten initial best-performing ones. This method is different from TuRBO which optimizes from a single, hand-selected configuration. We observed that this method performs better even when considering the plain TuRBO algorithm. In SAASBO, hyperparameter configurations are obtained using the NUTS sampler. They also sample multiple configurations.

---

> > ### Comment · Reviewer_qmcj · 2022-08-06
> > **Thanks for the response**
> >
> > Thanks for the response. My view of the paper remains unchanged but I certainly feel more confident about the score now.
> >
> > > we mean a uniform probability distribution in the stated range of hyperparameters.
> >
> > Perhaps explain this in the camera ready.
> >
> > > applicable benchmarks in MuJoCo to include
> >
> > LaMCTS and their modified TurBO can be found here: https://github.com/facebookresearch/LaMCTS/tree/main/LA-MCTS

---

### Official Review · Reviewer_4wPG · 2022-07-18

**Rating:** 7
**Confidence:** 3
**Soundness:** 3 good
**Presentation:** 3 good
**Contribution:** 3 good

**Summary:**

The paper proposes to apply a simple adaptive dimensional reduction technique on Trust Region Bayesian Optimization (TuRBO) for high dimensional Bayesian optimization task which has low effective dimensionality. The algorithm improves the probability of containing the optimum in the embedding space over previous work Hashing-enhanced Subspace BO (HESBO) by enforcing the hashing from the input dimension to the low-dimensional embedding to be evenly distributed. The paper demonstrates the efficacy of the proposed algorithm with a comprehensive empirical study.

**Questions:**

1. I might have missed the part. How many independent trails are conducted for each algorithm during experiment?
2. Could you provide results on the 60D Rover trajectory planning task that has been tested in TuRBO original experiment? I believe the task is available here https://github.com/zi-w/Ensemble-Bayesian-Optimization.


**Limitations:**

Not applied.

**Strengths And Weaknesses:**

- ***Strengths***:
1. Well-motivated algorithm design with the theoretical guarantee of its optimality over the existing HESBO in terms of probability of containing the optimal candidate in its embedding.
2. The proposed embedding is well-integrated into the existing HDBO framework TuRBO with appropriate algorithmic modification. Ablation studies prove the efficacy of each part of the proposed algorithm. Theoretical analysis in the appendix also shows that it inherits TuRBO's convergency.

- ***Weaknesses***:
1. Some of the definitions and concepts are confusing. In lines 59-62, the definition of the active subspace seems to be incomplete.  If for all x, z = f(x), g(z) = z, then the effective dimensionality could always be 1 regardless of the structure of X. I suggest defining the subspace as in definition 1 of the REMBO paper which defines the effective subspace and effective dimensionality. For definition in lines 157 - 162, could the author specify the reference? The classical definition of count-sketch structure assumes the pair-wise independence of the hashing functions [Charikar et al. 2002]. This is how HESBO implements its dimension reduction.

2. The classical count-sketch techniques rely on the pair-wise independencies of the hashing functions assigning input dimensions to different bins and the four-wise independencies of the sign assignment function as mentioned. The HESBO chooses the uniform random hashing which follows the assumption. The independency of hashing seems to be violated when BAxUS enforces that the input dimensions be evenly distributed to bins. Therefore, the original theoretical guarantee of the count-sketch structure probably does not hold in this case. The paper doesn't inherit the independency assumption on its count-sketch definition and lacks a discussion on this violation of independency of the hashing function.
In practice the violation of independency might not hurt much as in expectation HESBO also distributes the input dimensions evenly to target dimensions and the downstream performance of the proposed embedding is strong. However, the missing independency downgrades the theoretical contribution of the paper due to its violation of classic count-sketch structure and lacks a discussion on it, especially when considering that the work is motivated to improve upon HESBO while doesn't inherit its advantage strictly.

3. As far as I know TuRBO doesn't work well sometimes when only one trust region is operating and that motivates the TuRBO-m which is the m-trust-region variant of TuRBO (TuRBO-1) in the original paper. BAxUS lacks discussion over this problem. I expect a brief discussion over its choice of m and the corresponding implementation.

4. The descriptions could be confusing as the notations are not well specified between line 138 and line 147. And deferring algorithm 2, which is a critical part of the algorithm, to the appendix might not be a good choice.

*Reference*:
Charikar, M., Chen, K. and Farach-Colton, M., 2002, July. Finding frequent items in data streams. In International Colloquium on Automata, Languages, and Programming (pp. 693-703). Springer, Berlin, Heidelberg.

---

> ### Author Response · Authors · 2022-07-30
> **Reply to Reviewer 4wPG**
>
> We would like to thank the reviewer for their thorough review and helpful comments. We are delighted that the reviewer appreciates the originality of the presented algorithmic design and the theoretical contributions of our work. We have addressed the reviewer's request for TuRBO-5 and the other comments below.
>
> "Definition of the active subspace."
>
> We admit that this level of generality is confusing and unnecessary at this point. We have revised this definition and now it follows the notation from Alebo, reference [36] in our paper.
>
> "Reference of Definitions lines 157 - 162."
>
> Both definitions are our own, we made this clearer in the paper. The count-sketch algorithm and a count-sketch type embedding are different. The latter is characterized by its property that input dimensions can be assigned to target dimensions.
>
> "Violation of pairwise independence of the hashing functions."
>
> We agree with the reviewer that a discussion of this point would be a good addition to the paper, so we have added clarification in the uploaded pdf. Our algorithm indeed violates the pairwise independence assumption in cases where d does not divide D. In that case, the probability of an input dimension being assigned to a large bin is different from the probability of being assigned to a small bin. However, we note that while the pairwise independence is important for the count-sketch algorithm to obtain an unbiased estimate of the count of items, we do not see a connection between this property of the count-sketch algorithm to the success probability of an embedding. In particular, our theoretical analysis is valid because it does not depend on these properties.
>
> "Relation of BAxUS to TuRBO with multiple trust regions (TuRBO-m)"
>
> We added a comparison to TuRBO with multiple trust regions in Section 4. TuRBO-m leads to an improved performance for some of the benchmarks, in particular for Lasso-High. For the SVM benchmark, only 1000 evaluations are shown for TuRBO-m, we will update this plot in the coming days. Overall, the qualitative behavior does not change. We will update this plot as soon as possible. We added a short discussion of how BAxUS deals with multiple local minima in Section 3.2.
>
> "Notations not well-specified in lines 138-147 and Algorithm 2 in the appendix"
>
> We improved the writing of that section to make it clearer. We are keeping Algorithm 2 in the appendix due to space constraints and are confident that the revised text is now self-explanatory.
>
> “How many independent trails are conducted for each algorithm during experiment?”
>
> We have run 20 repetitions per experiment. We will state this in the paper.
>
> “Results on the 60D Rover?”
>
> The search space of Rover corresponds to the (x,y) coordinates of 30 waypoints. Thus, in order to search a lower-dimensional subspace, one would simply reduce the number of waypoints rather than applying a random embedding. Thus, Rover is not a reasonable benchmark to showcase the benefits of subspace-based approaches.

---

> > ### Comment · Reviewer_4wPG · 2022-08-09
> > **Reply to Authors**
> >
> > I appreciate the detailed response from the author and I believe most of my concerns are addressed. Here are my remaining questions about the count-sketch terminology and the corresponding independency.
> >
> > The count-sketch algorithm not only obtains an unbiased estimate of the count of items but also constructs a subspace embedding with a certain probability (Clarkson & Woodruff 2017). According to the HESBO paper, such subspace embedding preserve the mean and variance of the Gaussian process with certain kernel choices. The facts motivate conducting BO on the subspace embedding generated by the count-sketch algorithm. Therefore the paper should carefully differentiate its terminology from the existing "count-sketch subspace embedding". Also, it indicates that the success rate on the embedding might not be everything desired for BO.
> >
> > ***Reference***
> > Clarkson, K.L. and Woodruff, D.P., 2017. Low-rank approximation and regression in input sparsity time. Journal of the ACM (JACM), 63(6), pp.1-45.

---

> > > ### Author Response · Authors · 2022-08-09
> > > **Reply to Reviewer 4wPG**
> > >
> > > Thank you for this interesting comment. We agree that it is not clear whether our embedding is also an epsilon-subspace embedding. While we want to point out that by our ablation study, our embedding gives a better empirical performance, it is unclear at this point whether we inherit the same guarantees as the HeSBO embedding regarding the posterior mean and covariance of the GP. This is something we want to study in the future. In case we cannot prove that our embedding is also an epsilon-subspace embedding by the camera-ready deadline, we will change the terminology for the camera-ready version and add a short discussion of this point.

---

> > > > ### Comment · Reviewer_4wPG · 2022-08-09
> > > > **Post Rebuttal Thoughts**
> > > >
> > > > Thanks for the reply. I also believe that both its relative minor difference from the strict uniform random hashing and good downstream performance make it a promising embedding. I'm good with the proposed revisions. I've updated the score.

---

### Meta-Review · Area_Chair_oijF · 2022-08-24

**Recommendation:** Accept
**Confidence:** Certain

**Metareview:**

New active-subspaces type approach for high dimensional blackbox optimization that works with a family of nested subspaces of increasing dimensionality, with some theoretical control over the failure risk. Overall well written and complete with convincing experiments that indicate better performance than popular approaches like CMA-ES/Random-search as well as several recent methods. Please consider reviewer feedback on clarity and presentation for the final set of revisions.

**Award:**

No

---

### Decision · Program_Chairs · 2022-09-14

Accept